# V-HUB: A VISUAL-CENTRIC HUMOR UNDERSTANDING BENCHMARK FOR VIDEO LLMS

## ABSTRACT

AI models capable of comprehending humor hold real-world promise—for example, enhancing engagement in human-machine interactions. To gauge and diagnose the capacity of multimodal large language models (MLLMs) for humor understanding, we introduce V-HUB, a novel visual-centric video humor understanding benchmark. V-HUB comprises a curated collection of minimally verbal short videos, sourced from classic silent films and online resources, and reflecting real-world scenarios where humor can be appreciated purely through visual cues. Each video clip is paired with rich annotations, including captions, descriptions, and explanations, supporting evaluation tasks like caption matching and humor explanation. To broaden its applicability, we further construct an open-ended video QA task, making it readily integrable into existing video understanding benchmarks. We evaluate a diverse set of MLLMs, from specialized Video-LLMs to versatile OmniLLMs that can process audio, covering both open-source and proprietary domains. The experimental results expose the difficulties MLLMs face in comprehending humor from visual cues alone. For example, all models exhibit a marked performance drop on *caption matching* when moving from text-based to video-based evaluation (without audio). Our findings also demonstrate that incorporating audio helps with video humor understanding, highlighting the informativeness of sound and the promise of integrating richer modalities for complex video understanding tasks.

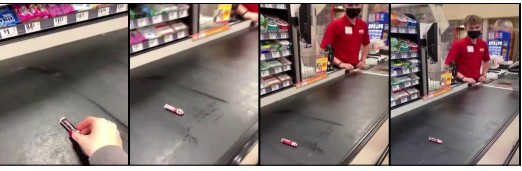

(a) **Visuals.** A man placed a battery on the conveyor belt, but it rolled against the belt's motion, forcing the cashier into an endless wait. For those who know the physics of a rolling cylinder on a moving conveyor, the scene feels even more clever.

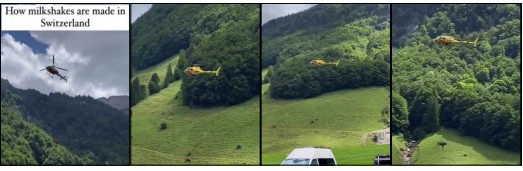

(b) **Visuals+Text.** The video shows an animal rescue, with a cow dangling beneath a helicopter, appearing to swirl midair. The scene seems routine at first, but the added text 'milkshakes' cleverly parallels the moment, making it unexpectedly witty.

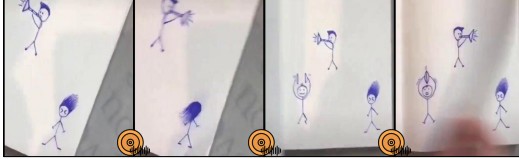

(c) **Visuals+Audio.** As the man flips through the pages, cartoon characters gradually appear, accompanied by a distinct melody. First, the dancer's rhythm and the suona player's piercing tune, then the cymbal player's resonant clash, together creating an evolving effect.

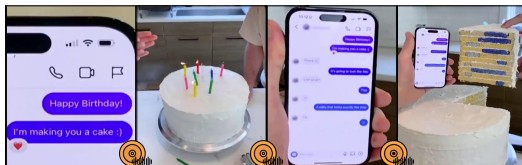

(d) **Visuals+Audio+Text.** A guy messaged his friend that he was making a birthday cake for them. After it was baked and sliced, the inside mimicked their chat bubble layout. The whole scene was made even merrier by the Happy Birthday melody.

Figure 1: Examples of visual-centric humor understanding, where 'audio' and 'text' refer to environmental sound (*cf.* human speech) and visual text, respectively.

# 1 INTRODUCTION

Humor enriches our daily lives and appears in many forms, from jokes and cartoons to comedies and viral videos. AI models capable of understanding humor hold promise for engaging with humans empathetically (Hampes, 2001; 2010), but perceiving and comprehending humor can be challenging even to humans due to the heavy reliance on nontrivial reasoning, social and cultural contexts, etc (see Figure 1). This, on the other hand, makes humor understanding a promising testbed to evaluate how well state-of-the-art AI models understand humor. Indeed, there has been a line of research centering around gauging the capability of pre-trained large language models (LLMs) for humor understanding (Hessel et al., 2022; Hyun et al., 2023; Ko et al., 2023), but parallel work on *multimodal* LLMs is still lacking, though they are more naturally suited for understanding multimodal humor.

In this work, we address this gap by investigating humor understanding with multimodal LLMs (MLLMs), focusing specifically on MLLMs that are capable of processing video. We choose video as the primary medium of humor, since it captures nuanced variations and diverse styles, presenting a unique challenge for MLLMs. For example, perceiving the humor in Figure 1d requires recognizing visual text and the layout of chat bubbles and understanding their temporal and semantic correspondences with the cut surface of the cake slice. While there have been a few benchmarks containing humorous videos (see Table 1), all of them were designed exclusively for the evaluation of LLMs (Ko et al., 2023; Hyun et al., 2023).[1] Moreover, they are limited in that each humor either is dominated solely by spoken language (Hyun et al., 2023) or can be understood only when both the video and linguistic cues are present (Ko et al., 2023), ignoring the fact that humans can understand humor from visual cues alone, exemplified by Charlie Chaplin's silent comedies.

To address this limitation, we curate a set of visual-centric humorous videos from two complementary sources: Charlie Chaplin's silent films and user-generated short funny videos. Silent film humor is conveyed through visual cues, but is thematically and culturally constrained due to the scripted performance. To increase diversity, we incorporate user-generated funny short videos from various occasions and cultural backgrounds. We rigorously filtered the videos to retain only those where the humor is primarily visual. Our final dataset consists of videos where humor is derived predominantly from the visual modality (e.g., 99% of all videos), making it visual-centric and more suitable for diagnosing the visual reasoning ability of MLLMs.

To assess how well MLLMs understand humor in video, we create a visual-centric humor understanding benchmark (v-HUB), which consists of three distinct tasks. (1) First, the *Caption Matching* task challenges MLLMs to align video captions with the corresponding videos. Apart from testing surface-level matching, the task is carefully designed to require an appreciation of nuanced, extended humor. (2) Second, the *Humor Explanation* task evaluates whether MLLMs can extract humor elements and provide accurate rationales. (3) Finally, the *Open-ended QA* task evaluate the MLLMs' fundamental understanding of videos from humor genre across temporal, descriptive, and causal dimensions, broadening the applicability of v-HUB. Together, these tasks provide a comprehensive framework to benchmark MLLMs in visual-centric humor understanding.

We evaluate representative MLLMs from both open- and closed-source domains. Depending on the input modalities, we consider the following three task settings. (1) The *Text-Only* setting assumes human-level interpretation of video contents and provides detailed human-written descriptions. (2) The *Video-Only* setting offers only videos (without audio) to assess the ability of MLLMs to derive humor solely from visual cues. (3) We further propose a novel *Video+Audio* setting that combines visual and auditory signals to determine whether sound cues—such as background music and sound effects—help MLLMs (aka. OmniLLMs) better understand humor.[2]

We empirically find that MLLMs generally perform better with text-only inputs than with video-only inputs (see Table 2). For example, Qwen2.5-VL-72B drops in accuracy from 0.719 to 0.673 on *Caption Matching*, and Gemini-2.5-Flash from 0.611 to 0.583, under the video-only setting, indicating their struggles in capturing subtle visual cues for humor understanding. Adding audio yields slight improvements across most OmniLLMs. For instance, MiniCPM-2.6-o improves from 0.364 to 0.404 in accuracy on *Caption Matching*, confirming the effectiveness of the audio modality, though it still lags behind the text-only setting. Overall, our v-HUB presents a new challenge and

---

[1]They translated videos into language descriptions and performed verbal humor evaluation with LLMs.

[2]In this work, audio primarily refers to environmental sound rather than human speech (see Section 2.2).

Table 1: Comparison between v-HUB and prior humor video datasets. Visual-Centric is a characteristic of data where the humor or content is derived predominantly from the visual modality.

| Dataset | Type | Visual-Centric | Tasks | | |
|---|---|---|---|---|---|
| | | | Explanation | Matching | Open-ended QA |
| NYCC (Hessel et al., 2022) | Cartoon | ❌ | ✅ | ✅ | ❌ |
| MUStARD (Castro et al., 2019) | Sitcom | ❌ | ❌ | ✅ | ❌ |
| WITS (Kumar et al., 2022) | Sitcom | ❌ | ❌ | ✅ | ❌ |
| UR-FUNNY (Hasan et al., 2019) | TED Talks | ❌ | ❌ | ✅ | ❌ |
| SMILE (Hyun et al., 2023) | Sitcom, TED Talks | ❌ | ✅ | ❌ | ❌ |
| ExFunTube (Ko et al., 2023) | Short videos | ❌ | ✅ | ❌ | ❌ |
| v-HUB (ours) | Short videos, Silent films | ✅ | ✅ | ✅ | ✅ |

contributes to a comprehensive evaluation of MLLMs. It exposes their weakness in visual-centric humor understanding, stresses the need for enhancing their visual reasoning capabilities, and highlight the promise of integrating additional modalities like sound for video understanding.

## 2 CURATING VISUAL-CENTRIC HUMOROUS VIDEOS

### 2.1 HUMOR VIDEO SOURCES

Our goal is to collect humorous videos that are visual-centric and illustrate diverse humor. A straightforward approach is to collect humorous clips from silent comedies that are entirely devoid of speech. Though silent films may contain recorded music, sound effect, and few captions, which may contribute to the expression of humor, the humor primarily arises from the visual modality. A major issue with silent film clips is that they have rather narrow themes and employ limited storytelling techniques. To enhance the diversity of humor in our dataset, we further incorporate user-generated short funny videos from the Internet. Specifically, we selected videos from an X account (@humansnocontext) that frequently shares humorous clips with minimal reliance on speech or text-based context. Thus, our dataset comprises humorous videos from two different domains that complement each other (see Figure 2):

- **Charlie Chaplin's Silent Films:** We reviewed Charlie Chaplin's classic silent films from 1914 to 1938 and collected 729 funny clips. Each humor is ensured to be self-contained, without relying on additional video contexts. Figure 2 shows an example in this domain.

- **User-Generated Funny Videos:** We reviewed the X user @humansnocontext's tweets posted between March 28, 2023 and October 12, 2024 and collected 18080 short funny videos.

### 2.2 PREPROCESSING AND FILTERING

We preprocess and filter the initially collected videos according to duration, appropriateness, and speech reliance, sequentially. (1) **Duration:** We retain videos ranging from 5 to 60 seconds long. Short clips under 5 seconds generally fail to convey meaningful humor, while clips exceeding 1 minute often rely on dialogue. For silent films, we segment long scenes to isolate individual humorous moments, ensuring that each segment captures the full humor, without becoming too long for generation tasks. (2) **Appropriateness:** To ensure that the contents of our videos are appropriate, we adhered to the safety objectives outlined in Thoppilan et al. (2022) and excluded videos that violated the established criteria (see details in Appendix B.1). (3) **Speech reliance:** We minimize reliance on speech. Since there is little to no speech in Charlie Chaplin's silent films, we primarily focused on user-generated funny videos and employed both manual and automatic approaches to filter out speech-heavy videos (see details in Appendix B.1).

### 2.3 ANNOTATION

We recruited eight annotators based on the following criteria: (1) sufficient English proficiency to understand video content, (2) broad cultural knowledge to interpret humor arising from various contexts, and (3) strong observational skills assessed through a qualification test (see Appendix B.2). To ensure consistency, we provided detailed guidelines for each annotation task and created a

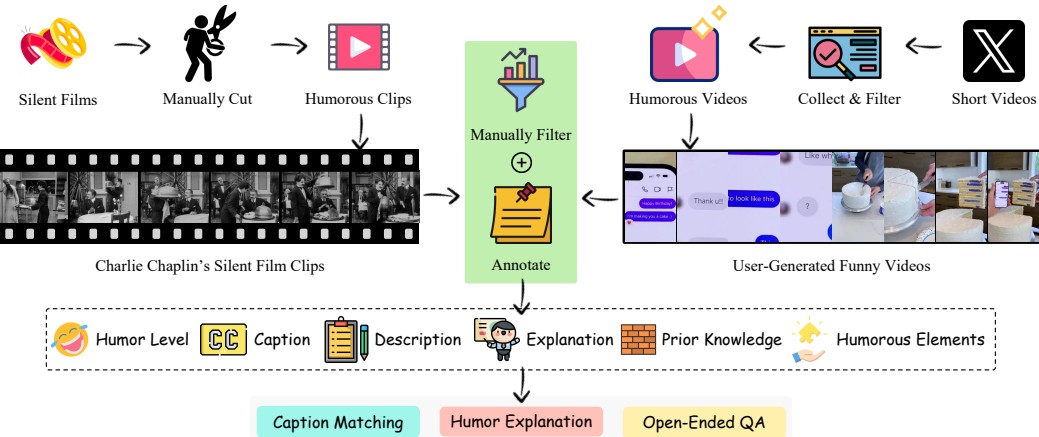

Figure 2: Data Curation Pipeline. To collect visual-centric humorous videos, the pipeline consists of two main stages: (a) *Humorous video collection*, where annotators identify timestamps of self-contained humorous clips for silent films and verify humor presence in short videos (see Section 2.1). (b) *Filtering and annotation*, where only visual-dominant humor is retained and annotated (see Section 2.3). The annotation is further used for task construction (see Section 3).

reference manual for on-demand use. Each video underwent three rounds of annotation to guarantee correctness and thoroughness. We conducted the following primary annotation tasks (see Figure 3 for an example annotation):

- **Humor Evaluation:** Annotators independently evaluated whether the video was humorous.

- **Captioning:** Each annotator was asked to write two types of captions for each video, without seeing existing annotations, including captions and descriptions, from other annotators, thus ensuring an independent and unbiased judgment.
  1. *Descriptive Captions* directly describe or highlight the humor present in the video content from the original publisher's perspective.
  2. *Creative Captions* extend beyond the video's original humor by adding imaginative or novel elements (see the added visual caption in Figure 1b).

  The dual-caption annotation supports a comprehensive assessment of humor in video from both comprehension and generation perspectives (see Section 3).

- **Video Description:** Annotators were instructed to describe the events in each video without making inferences, focusing only on observable objects, actions, and expressions. After the first annotator completes the video description, subsequent annotators review and refine these descriptions for correctness and completeness.

- **Video Labeling:** Annotators labeled the key humor sources (e.g., human actions, objects, visual effects, or sound cues) in each video and noted whether any visual text was present. If an element appeared, but did not contribute to humor, it was not selected.

- **Humor Explanation:** Three annotators sequentially create and refine humor explanations by adding missing details, guaranteeing comprehensive coverage of the labeled humor sources through an iterative refinement process.

## 2.4 DATA ANALYSIS

After all filtering processes, we were left with 960 videos, including 267 silent humor clips from Charlie Chaplin and 693 user-generated short funny videos from the Internet. The total duration of the videos is 4h, and the average duration is around 15s. All of them rely on the visual modality to express humor. We identify two key modalities that dominate the delivery of humor: visuals and audio. Apart from 600 videos (63%) conveying humor primarily via pure visual cues (denoted by 'Visual'), 92 videos (9%) contain additional linguistic cues in visual form—such as embedded captions and subtitles (denoted by 'Visual+Text')—that extend humor, 214 video humor (22%) is enhanced by

Figure 3: Example annotation of a short video that conveys humor through visuals, visual text, and background sound. Knowing the Happy Birthday melody makes the video merrier (see Section 2.3).

additional sound that covers non-speech auditory elements, such as background music, sound effects, and character vocalizations (denoted by 'Visual+Audio'), and 46 videos (5%) convey humor through visuals, sound, and visual text (denoted by 'Visual+Audio+Text'). The video distribution over the four groups is illustrate in Figure 8 in Appendix C, and more analysis can be found there.

# 3 V-HUB: A VISUAL-CENTRIC HUMOR UNDERSTANDING BENCHMARK

## 3.1 EVALUATION TASKS

To comprehensively evaluate the capability of MLLMs in humor understanding, we propose three tasks that reflect different aspects of humor reasoning: Caption Matching, Humor Explanation, and Open-ended QA:

- **Caption Matching.** In this discriminative task, models must correctly associate videos with their corresponding captions. Unlike ordinary caption matching tasks, our design challenges MLLMs to go beyond surface-level matching and assess their ability to understand video humor that is pronounced by *creative captions* from a generation perspective. For each video with a creative caption, we randomly sample four *descriptive captions* from other videos as the distractors.

- **Humor Explanation.** In this generative task, models must identify humor points within each video, provide coherent explanations, and reference relevant visual or auditory cues.

- **Open-ended QA.** To further assess the fundamental understanding of video content, we generate a set of open-ended question-answer pairs for each video (see details in Appendix D.1). These questions—automatically generated by GPT-4o (Hurst et al., 2024) and manually verified—encompass temporal, descriptive, and causal aspects (Xiao et al., 2021).[3] This extends the benchmark beyond humor-specific reasoning, providing a broader assessment of video reasoning skills.

## 3.2 EVALUATION METHODS

We employ different evaluation strategies depending on the task type:

- **Accuracy.** For the caption matching task, we measure accuracy to determine whether the model correctly identifies the most appropriate response.

- **Quality of Open-ended Responses.** For humor explanation and open-ended QA tasks, we adopt both automatic and human evaluation approaches:
  - *Semantic Similarity.* We compute similarity scores between model-generated answers and human-provided answers using BERTScore (Zhang* et al., 2020), which captures fine-grained

---

[3]There are 62, 675, and 223 QA pairs for temporal, descriptive, and causal questions, respectively.

semantic similarity beyond simple word overlap. In addition, we employ SentBERT (Reimers & Gurevych, 2019) to assess sentence-level semantic coherence, as well as METEOR (Banerjee & Lavie, 2005), which provides a more nuanced assessment of semantic adequacy and fluency.

– AutoDQ (Wang et al., 2024a): This method evaluates the presence of humor-related events in the generated explanations. AutoDQ extracts key events from the model's output and compares them to ground truth (GT) annotations using entailment analysis. It provides three metrics: recall, precision, and F1 score (see Appendix D.2 for details).

– *Human Evaluation*. We randomly sample a subset of model-generated explanations and compare them with human-written explanations. The evaluators rate the explanations based on accuracy and logicality, providing insight into the gap between human and MLLMs' explanations. Results are presented in Table 9 in Appendix E.

## 4 EXPERIMENTS

### 4.1 EXPERIMENTAL SETUP

**MLLMs.**   We consider both proprietary and public MLLMs like Gemini-2.5-Flash (Team et al., 2025) and Qwen2.5-VL (Bai et al., 2025). OmniLLMs such as Video-SALMONN-2 (Tang et al., 2025) and Qwen-2.5-Omni (Xu et al., 2025), which can process audio, are also included (an overview of all evaluated MLLMs is presented in Table 8).

**Evaluation settings.**   To understand the roles of different modalities in video humor understanding, we consider the following three settings: Text-Only, Video-Only, and Video+Audio, which means models are tested with text, video (w/ audio), and video-audio inputs, respectively.

• *Text-Only*. In this setting, models receive detailed human-written video descriptions; no visual or audio information is available to the models. Thus, it evaluates the language reasoning ability of MLLMs in isolation.

• *Video-Only*. Models are provided with only raw video frames, without audio. This setting assesses their intrinsic visual comprehension capabilities. Depending on the presence of visual text, we further divide results into two groups: 'w/ visual text' and 'w/o visual text'.

• *Video+Audio*.  Models receive both video frames and audio signals, allowing us to examine whether the inclusion of auditory information improves humor understanding. Depending on the contribution of audio to humor, we further divide results into two groups: 'w/ humor audio' and 'w/o humor video'.

### 4.2 MAIN RESULTS

Based on the results in Table 2, we analyze the humor competence of MLLMs along three dimensions: video humor discovery, understanding, and subtle humor inference.  Our results reveal several shortcomings of MLLMs: they (i) struggle to identify humorous elements when explicit cues are absent, (ii) inadequately fuse information across modalities for understanding, and (iii) show limited capacity for inferring subtle humor.

**Limited ability in humor discovery.**   Across settings, models consistently perform better on Open-ended QA than on Humor Explanation. This performance disparity reveals that they are limited in perceiving humor. For example, in the Text-Only setting, Qwen-2.5-VL-72B, whose score drops from 0.792 in QA to 0.553 in Humor Explanation. These findings suggest that models are more successful when the question itself provides explicit cues that direct attention to a specific humorous element in the scene. By contrast, the Humor Explanation task, which requires models to independently identify and articulate the source of humor without such guidance, poses a greater challenge. This indicates that while MLLMs are often able to reason about humor once it is highlighted for them, they struggle with the more cognitively demanding task of discovering humor directly from contextual cues.

**Heavy reliance on linguistic cues for humor understanding.**   Comparing text-based and video-based evaluations, we observe marked differences across all three tasks, where the Text-Only setting yields substantially higher scores than the video-based settings, implying that current MLLMs are

Table 2: Model performance on Humor Explanation, Caption Matching, and Open-ended QA.

| MLLMs | Explanation | | | | Matching | Open-ended QA | |
|---|---|---|---|---|---|---|---|
| | SentBERT | METEOR | BERTScore | AutoDQ | Accuracy | SentBERT | METEOR |
| *Text-Only* | | | | | | | |
| Gemini-2.5-flash | 0.556 | 0.248 | 0.580 | 0.319 | 0.611 | 0.748 | 0.652 |
| Video-SALMONN-2 | 0.575 | 0.242 | 0.589 | 0.292 | 0.367 | 0.602 | 0.445 |
| MiniCPM2.6-o | 0.558 | 0.239 | 0.562 | 0.298 | 0.518 | 0.578 | 0.467 |
| Qwen-2.5-Omini | 0.547 | 0.232 | 0.570 | 0.295 | 0.644 | 0.740 | 0.555 |
| Qwen-2.5-VL-72B | 0.553 | 0.250 | 0.578 | 0.324 | 0.719 | 0.792 | 0.622 |
| Intern3.5-VL | 0.567 | 0.255 | 0.580 | 0.326 | 0.632 | 0.721 | 0.578 |
| GPT-4o | 0.569 | 0.256 | 0.581 | 0.350 | 0.767 | 0.720 | 0.666 |
| *Video-Only* | | | | | | | |
| Gemini-2.5-flash | 0.469 | 0.200 | 0.550 | 0.163 | 0.583 | 0.434 | 0.275 |
| video-SALMONN-2 | 0.281 | 0.150 | 0.504 | 0.043 | 0.259 | 0.311 | 0.160 |
| MiniCPM2.6-o | 0.387 | 0.164 | 0.520 | 0.100 | 0.364 | 0.323 | 0.110 |
| Qwen-2.5-Omini | 0.388 | 0.157 | 0.521 | 0.144 | 0.55 | 0.385 | 0.104 |
| Qwen-2.5-VL-72B | 0.452 | 0.188 | 0.547 | 0.135 | 0.673 | 0.459 | 0.201 |
| Intern3.5-VL | 0.433 | 0.186 | 0.543 | 0.117 | 0.640 | 0.393 | 0.230 |
| GPT-4o | 0.478 | 0.198 | 0.547 | 0.198 | 0.667 | 0.431 | 0.300 |
| Video+Audio | | | | | | | |
| Gemini-2.5-flash | 0.472 | 0.200 | 0.550 | 0.165 | 0.588 | 0.428 | 0.275 |
| video-SALMONN-2 | 0.296 | 0.176 | 0.506 | 0.055 | 0.255 | 0.319 | 0.180 |
| MiniCPM2.6-o | 0.419 | 0.176 | 0.523 | 0.110 | 0.404 | 0.348 | 0.245 |
| Qwen-2.5-Omini | 0.442 | 0.177 | 0.531 | 0.120 | 0.623 | 0.439 | 0.164 |

heavily dependent on linguistic cues for humor understanding. For example, On Open-ended QA, Qwen-2.5-VL-72B achieves a SentBERT score of 0.792 with text input, but it plummets to 0.459 when presented with raw video (w/o audio). While the addition of audio provides a marginal but consistent performance boost, this gain is minimal compared to the contribution of text. This wide performance gap suggests that MLLMs' cross-modal fusion capabilities are still underdeveloped, leading them to rely predominantly on linguistic cues rather than effectively integrating visual and auditory signals. Thus, future work is well-suited for enhancing MLLMs beyond language understanding.

**Incapability for subtle humor inference.** The Caption Matching task goes beyond surface-level linking between literal descriptions and videos; instead, it requires models to find the *creative caption* that enhances or extends humor in the video. We find that most models exhibit limited performance (e.g., below 0.8), suggesting their incompetence for subtle humor inference. For example, under the most favorable conditions, that is, in the Text-Only setting, the top-performing model, GPT-4o, achieves an accuracy of only 0.767. The difficulty is magnified when models must process raw video data. For example, video-SALMONN-2's accuracy falls sharply from 0.367 in the Text-Only setting to 0.255 in the Video+Audio condition. This pronounced struggle to connect creative, non-obvious text to original visual humor context reveals a critical weakness in the models' capacity for the abstract, implicit cross-modal reasoning that is fundamental to comprehending sophisticated humor.

## 4.3 FURTHER ANALYSIS

To conduct a deeper analysis of model results, we further divide our experimental results based on previously annotated humor modalities and background knowledge essential for delivering humor in video, to analyze how different types of humor affect models' explanatory capability.

**Both audio and visual text help with humor understanding.** As shown in Table 3, MLLMs perform better on videos containing visual text or subtitles than on those without linguistic cues under the Video+Audio setting. For example, Gemini-2.5-Flash attains a SentBERT score of 0.532 and a METEOR score of 0.212 for humor explanation with visual text, compared to 0.486 and 0.202 without visual text. When sound does not contribute to humor, the advantage of visual text becomes even more pronounced: Gemini-2.5-Flash improves from 0.455 to 0.509 in Explanation SentBERT and from 0.215 to 0.739 in Matching Accuracy with visual text. These results indicate that while

Table 3: The impacts of audio (i.e., sound) and visual text on video humor understanding.

| Models | Sound contributing to humor | | | | | Sound not contributing to humor | | | | |
|---|---|---|---|---|---|---|---|---|---|---|
| | Explanation | | Matching | Open-ended QA | | Explanation | | Matching | Open-ended QA | |
| | SentBERT | METEOR | Accuracy | SentBERT | METEOR | SentBERT | METEOR | Accuracy | SentBERT | METEOR |
| *w/ visual text* | | | | | | | | | | |
| Gemini-2.5-flash | 0.532 | 0.212 | 0.630 | 0.488 | 0.359 | 0.509 | 0.219 | 0.739 | 0.476 | 0.319 |
| video-SALMONN-2 | 0.292 | 0.190 | 0.261 | 0.319 | 0.189 | 0.280 | 0.176 | 0.293 | 0.318 | 0.189 |
| MiniCPM2.6-o | 0.490 | 0.189 | 0.348 | 0.374 | 0.289 | 0.474 | 0.197 | 0.489 | 0.380 | 0.282 |
| Qwen-2.5-Omni | 0.512 | 0.192 | 0.783 | 0.525 | 0.176 | 0.467 | 0.190 | 0.75 | 0.453 | 0.166 |
| *w/o visual text* | | | | | | | | | | |
| Gemini-2.5-flash | 0.486 | 0.202 | 0.523 | 0.409 | 0.265 | 0.455 | 0.194 | 0.215 | 0.423 | 0.266 |
| video-SALMONN-2 | 0.296 | 0.181 | 0.243 | 0.299 | 0.168 | 0.298 | 0.173 | 0.178 | 0.325 | 0.182 |
| MiniCPM2.6-o | 0.451 | 0.178 | 0.341 | 0.355 | 0.271 | 0.393 | 0.170 | 0.215 | 0.334 | 0.226 |
| Qwen-2.5-Omni | 0.471 | 0.176 | 0.551 | 0.457 | 0.150 | 0.422 | 0.173 | 0.197 | 0.422 | 0.166 |

Table 4: The impact of requiring background knowledge support on video humor understanding.

| Models | Explanation | | | | Matching | Open-ended QA | |
|---|---|---|---|---|---|---|---|
| | SentBERT | METEOR | BERTScore | AutoDQ | Accuracy | SentBERT | METEOR |
| Gemini-2.5-flash | 0.503 | 0.211 | 0.568 | 0.194 | 0.633 | 0.437 | 0.268 |
| video-SALMONN-2 | 0.273 | 0.153 | 0.513 | 0.139 | 0.266 | 0.306 | 0.159 |
| MiniCPM2.6-o | 0.404 | 0.166 | 0.532 | 0.093 | 0.378 | 0.318 | 0.103 |
| Qwen-2.5-Omni | 0.404 | 0.157 | 0.531 | 0.026 | 0.571 | 0.384 | 0.105 |

both audio and visual text help with humor understanding, MLLMs rely more heavily on textual cues, and the presence of *visual text can effectively compensate for the absence of informative sound*.

**Background knowledge does not necessarily improve video humor understanding.** The results in Table 2 and Table 4 show that there is no significant difference between the mean scores and the scores for videos that need background knowledge to perceive humor under Video-Only setting. For example, under the Video-Only setting, the Gemini-2.5-flash attains an average BERTScore of on the Explanation task for Background-Dependent videos 0.568, which is statistically similar to its BERTScore of 0.550 on the full dataset. This suggests that the language-model component of MLLMs already encodes most of the cultural background knowledge necessary for humor comprehension, meaning that *the absence of explicit background knowledge in the input does not significantly degrade their performance*. MLLMs do not show a significant disadvantage in understanding videos that require background knowledge compared to those that do not, potentially because video humor rarely relies on specific background knowledge, making it universally understandable.

**Knowledge-based cues facilitate humor understanding.** We identified 357 videos that require contextual background knowledge and evaluated MLLMs under two settings: with and without the explicit provision of such knowledge. As shown in Table 5, MLLMs consistently achieve higher performance when background knowledge is provided. For instance, Qwen-2.5-Omni attains a SentBERT score of 0.514 and an Explanation BERTScore of 0.557 with background knowledge,

Table 5: The impact of background knowledge on video humor understanding.

| MLLMs | Explanation | | | | Matching | Open-ended QA | |
|---|---|---|---|---|---|---|---|
| | SentBERT | METEOR | BERTScore | AutoDQ | Accuracy | SentBERT | METEOR |
| *w/ Background Knowledge* | | | | | | | |
| video-SALMONN-2 | 0.468 | 0.173 | 0.563 | 0.112 | 0.331 | 0.397 | 0.195 |
| MiniCPM-2.6-o | 0.518 | 0.203 | 0.557 | 0.191 | 0.445 | 0.430 | 0.204 |
| Qwen-2.5-Omni | 0.514 | 0.197 | 0.557 | 0.177 | 0.667 | 0.496 | 0.220 |
| *w/o Background Knowledge* | | | | | | | |
| video-SALMONN-2 | 0.287 | 0.178 | 0.515 | 0.023 | 0.261 | 0.300 | 0.174 |
| MiniCPM-2.6-o | 0.444 | 0.181 | 0.540 | 0.113 | 0.412 | 0.351 | 0.252 |
| Qwen-2.5-Omni | 0.462 | 0.181 | 0.545 | 0.129 | 0.630 | 0.445 | 0.158 |

Table 6: The impact of video era on video humor understanding.

| MLLMs | Explanation | | | | Matching | Open-ended QA | |
|---|---|---|---|---|---|---|---|
| | SentBERT | METEOR | BERTScore | AutoDQ | Accuracy | SentBERT | METEOR |
| *Last-Century Charlie Chaplin's Silent Films* | | | | | | | |
| Gemini-2.5-flash | 0.422 | 0.188 | 0.541 | 0.130 | 0.562 | 0.386 | 0.221 |
| video-SALMONN-2 | 0.281 | 0.146 | 0.509 | 0.012 | 0.165 | 0.296 | 0.154 |
| MiniCPM2.6-o | 0.343 | 0.150 | 0.508 | 0.097 | 0.307 | 0.314 | 0.128 |
| Qwen-2.5-Omni | 0.339 | 0.144 | 0.510 | 0.096 | 0.494 | 0.337 | 0.119 |
| *Contemporary User-Generated Funny Video* | | | | | | | |
| Gemini-2.5-flash | 0.487 | 0.205 | 0.553 | 0.185 | 0.592 | 0.452 | 0.295 |
| video-SALMONN-2 | 0.280 | 0.151 | 0.503 | 0.051 | 0.296 | 0.315 | 0.163 |
| MiniCPM2.6-o | 0.404 | 0.170 | 0.524 | 0.103 | 0.385 | 0.326 | 0.104 |
| Qwen-2.5-Omni | 0.407 | 0.162 | 0.525 | 0.174 | 0.571 | 0.404 | 0.098 |

Table 7: Comparison between MLLMs and their base LLMs under the Text-Only setting.

| Models | Open-ended QA | | |
|---|---|---|---|
| | SentBERT | METEOR | BERTScore |
| Qwen2.5-VL-72B | 0.792 | 0.622 | 0.738 |
| Qwen2.5-72B | 0.710 | 0.636 | 0.700 |
| Qwen2.5-Omni-7B | 0.740 | 0.555 | 0.698 |
| Qwen2.5-7B | 0.692 | 0.539 | 0.667 |

compared to 0.462 and 0.545 without. These findings suggest that while MLLMs implicitly encode certain aspects of cultural context, their *comprehension of humor is significantly enhanced by the explicit provision of background knowledge*, underscoring the central role of linguistic and knowledge-based cues in complex video humor understanding tasks.

**MLLMs have greater difficulty in comprehending humor in historically distant videos.** We analyze the performance of MLLMs under the Video-Only setting across two subsets from distinct eras: last-century Charlie Chaplin's silent films (CCSF) and contemporary user-generated funny videos (UGFV). As shown in Table 6, MLLMs consistently achieve higher scores on UGFV across all evaluation metrics. For example, Gemini-2.5-flash attains a BERTScore of 0.553 for Humor Explanation and 0.560 for Open-ended QA on UGFV videos, compared to 0.541 and 0.545, respectively, on CCSF videos. These findings suggest that MLLMs face greater difficulty in comprehending humor in historically distant videos, *highlighting the sensitivity of humor understanding to the temporal and cultural context of videos*.

**MLLMs vs. their base LLMs.** MLLMs are usually derived from a pre-trained base LLM by adding a visual encoder or multimodal modules. For instance, Qwen2.5-VL-72B extends Qwen2.5-72B, and Qwen2.5-Omni extends Qwen2.5-7B (see Table 7). In the Text-Only setup, Qwen2.5-Omni surpasses Qwen2.5-7B with a SentBERT score of 0.740 (vs. 0.692) and a BERTScore score of 0.698 (vs. 0.667) on Open-ended QA task, suggesting that *multimodal training can confer advantages even when only textual descriptions are available, possibly because the model has learned richer contextual associations during training*. Please refer to Table 12 for more details on humor explanation and caption matching tasks.

## 5 RELATED WORK

**Video LLMs.** Video LLMs have shown remarkable performance in many traditional video processing tasks such as video captioning (Xu et al., 2016; Agrawal et al., 2019; Plummer et al., 2017), video question answering (Antol et al., 2015; Xiao et al., 2021; Yu et al., 2019; Fu et al., 2025), and grounding (Kazemzadeh et al., 2014; Wu et al., 2022). However, most existing benchmarks primarily

target general video understanding tasks, such as MVBench (Li et al., 2024), Video-MME (Fu et al., 2025), PerceptionTest (Patraucean et al., 2023), MLVU (Zhou et al., 2025), and LVBench (Wang et al., 2024c), which mainly assess the recognition of basic visual cues across videos of varying lengths. Others are designed to evaluate specific video understanding capabilities, including temporal grounding (Gao et al., 2017; Lei et al., 2021; Hendricks et al., 2017; Wang et al., 2024d), video object detection (Shang et al., 2019; 2017), and video hallucination (Wang et al., 2024e; Leng et al., 2024). But there remains a pressing need for benchmarks that evaluate higher-level cognitive abilities, such as social intelligence, in order to better measure the gap between human and MLLMs' performance.

Our work narrows this gap. We expand the evaluation spectrum of video LLMs by introducing a novel humor understanding evaluation framework, formulating a humor generation task, and presenting a first comprehensive evaluation.

**Humor Video Understanding.** Humor understanding has been a popular research topic in the area of artificial intelligence and has its roots in cognitive science (Hampes, 2001; 2010),. Early works focus on verbal humor in the form of jokes, sarcasm, etc (Chłopicki, 2005; Petrović & Matthews, 2013; Joshi et al., 2017). As AI models become capable of processing more data modalities like images, videos, and audio, many efforts have been devoted to multimodal humor understanding. For example, Hessel et al. (2022) analyze cartoon images and humorous captions, Desai et al. (2022); Kumar et al. (2022) investigate sarcasm, a special humor type, with image-language data, and Castro et al. (2019); Hasan et al. (2019); Kayatani et al. (2021); Patro et al. (2021); Alnajjar et al. (2022) focus on laughter detection and explanation with videos, including video presentations, sitcoms, and stand-up comedy, but laughter is not the only emotion reaction to humor.

With the development of LLMs, recent works tested how well LLMs understand multimodal humor with image-language and video-language humor (Hessel et al., 2022; Alnajjar et al., 2022), but a parallel evaluation of video LLMs is still missing. While previous works have introduced several video-based humor datasets (Kumar et al., 2022), humor in their videos is either primarily dominated by spoken dialogue or restricted to those that have to be understood relying on both visual and linguistic cues. In contrast, we introduce a visual-centric humor video dataset designed to simulate common scenarios where humans can understand humor purely from visual cues.

Going beyond visual and verbal humor understanding, sound has been found informative of commonsense (Zhao et al., 2022; Zellers et al., 2022), and several studies demonstrate that integrating textual, acoustic and visual characteristics can significantly improve humor detection accuracy (Chandrasekaran et al., 2016; Hasan et al., 2019). Since multimodal LLMs have recently been extended to support audio processing (aka. OmniLLMs), we propose and conduct a first evaluation of MLLMs on video humor understanding that involves sound.

## 6 CONCLUSION

We have introduced v-HUB, a visual-centric humor understanding benchmark. v-HUB is designed to assess and diagnose the capability of MLLMs for video humor understanding. It contains a collection of funny videos collected from two complementary domains. Each clip is annotated with captions, descriptions, explanations, etc., supporting evaluation tasks such as caption matching and humor explanation. To broaden the applicability of v-HUB, we further construct an open-ended task, contributing to a comprehensive evaluation of MLLMs for video understanding. We evaluated a diverse range of MLLMs, spanning open-sourced and proprietary domains and covering specialized video LLMs and versatile OmniLLMs. Our findings reveal that current MLLMs heavily rely on linguistic cues for humor understanding, but are weak in deriving nuanced visual cues for understanding sophisticated video humor. Moreover, we empirically find that including audio is helpful for humor understanding, highlighting the informativeness of sound and the promise of incorporating rich modalities for complex video reasoning tasks.

ETHICS STATEMENT

Our work follows widely recognized ethical principles in computing research, including the ACM Code of Ethics and the ICLR ethics guidelines. In developing our benchmark, we considered the following aspects:

- **Contribute to Society and to Human Well-being:** Our dataset is intended to advance multimodal AI research on humor understanding, a capability that has broad applications in safe content moderation, assistive technologies and cross-cultural communication. We employed three Human Intelligence Tasks (HITs) to gather data, and we fully considered cultural and linguistic diversity in humor to minimize potential negative impacts as much as possible, such as reinforcing harmful stereotypes, exposing sensitive content, or infringing on personal safety and privacy. To ensure accessibility and inclusivity, the dataset will be made broadly available for non-commercial research purposes.

- **Uphold High Standards of Scientific Excellence:** We consistently adhere to principles of transparency and rigor throughout dataset curation and analysis. We meticulously document all preprocessing, annotation and evaluation procedures, and we will publicly release the relevant code to ensure independent verification and reproducibility. Furthermore, we provided fair compensation to all annotation personnel, ensuring their hourly wages exceeded the local minimum wage. Finally, we did not fabricate, falsify, or misrepresent data. Beyond the data annotators, no other human subjects were directly involved, and no personally identifiable information was used. Therefore, no additional ethics approval is required.

- **Avoid Harm:** We carefully considered potential risks that could arise from constructing and releasing a humor video benchmark. To mitigate negative consequences, we implemented a multi-stage screening mechanism to exclude humorous content featuring violence, discrimination, or relying on stereotypes targeting vulnerable groups. Simultaneously, we adopted a three-person collaborative annotation scheme, ensuring that each data entry underwent three rounds of annotation. We explicitly document the limitations of the dataset to minimize unintended harm arising from cross-cultural or linguistic misunderstandings. Furthermore, we analyzed potential downstream risks, such as misuse for generating harmful or offensive humor, and explicitly warn against such applications in the dataset license.

- **Be Honest, Trustworthy and Transparent:** We transparently disclose the characteristics, strengths, and limitations of the dataset. While this dataset thoroughly considers cultural and linguistic diversity in humor and encompasses diverse humor scenarios, it cannot cover all dimensions of global humor cultures. Furthermore, Charlie Chaplin's Silent Films constitute a significant portion of the dataset, inevitably introducing potential cultural bias. We confirm that there are no conflicts of interest that may compromise the independence of our research, and all funding sources are clearly acknowledged. At the same time, we guarantee that we do not misrepresent related work, nor do we claim capabilities beyond what our benchmark enables.

- **Be Fair and Take Action not to Discriminate:** We strived for fairness in both dataset curation and evaluation. And we made efforts to avoid humor that demeans particular groups. We also emphasize that the dataset should not be used to develop systems that discriminate, disenfranchise, or oppress individuals.

- **Respect the Work Required to Produce New Ideas and Artefacts:** We credit the creators of ideas, inventions, work, and artefacts, and respect copyrights and property. All videos are sourced from publicly available materials with appropriate licenses. Where possible, we provide attribution to content creators and respect cultural heritage by excluding sensitive or protected media. Where possible, we will provide attribution to content creators and respect their work by excluding sensitive or protected media content.

- **Respect Privacy:** Our work did not use private or personally identifiable data. All videos have been anonymized or utilize publicly available Charlie Chaplin silent films, carefully mitigating the risk of re-identification. Furthermore, the dataset is restricted to legitimate academic research under the dataset license.

- **Honour Confidentiality:** Our work did not involve any confidential or proprietary information. Reviewers and collaborators were only provided with materials approved for release. We commit to maintaining confidentiality in peer review and in handling sensitive communications.

## REPRODUCIBILITY STATEMENT

We have taken several steps to ensure the reproducibility of our work. The details of dataset collection, filtering, and annotation protocols are described in Section 2 of the main paper, with further implementation details provided in Appendix B. The evaluation metrics and experimental setups for all baseline models are reported in Section 4 and Appendix D. We also provide the full benchmark dataset along with evaluation scripts to allow replication of our results. All hyperparameters and experimental configurations are listed in the supplementary materials. Due to the use of API-based models and inherent randomness (e.g., random seeds during evaluation), reproduced results may exhibit slight variations from those reported, but overall trends and conclusions remain consistent.

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

## A ADDITIONAL RELATED WORK

### A.1 FROM LLMS TO VIDEO LLMS

Large language models have demonstrated outstanding capabilities in many domains, including natural language processing, coding, math, and reasoning, ushering in new breakthroughs for video understanding technology. Video LLMs integrate visual encoders with LLMs, leading to a unified model to reason across video and language in the same language space (Wang et al., 2024b; Liu et al., 2024; Lin et al., 2023). Early video LLMs employ pre-trained image encoder and video encoder to encode only video frames (Zhang et al., 2023; Maaz et al., 2023; Li et al., 2023; Lin et al., 2023). Recent works augment video LLMs with an audio encoder to align visual, auditory, and textual modalities in the same language space. Moreover, the audio encoder is supposed to capture diverse environmental sound apart from human speech since sound has been shown to contain amounts of commonsense knowledge (Cheng et al., 2024; Xu et al., 2025).

## B CROWDWORKING DETAILS

### B.1 PROCESSING AND FILTERING

**Harmful content detection.** Before the annotation process began, we manually filtered out videos that contained potentially harmful content to ensure the video data's safety and quality (Figure 4 visualizes our annotation interface). Based on the criteria outlined by Thoppilan et al. (2022), we defined 6 categories of harmful contents, following aspects are checked for each video.

- *Discrimination.* Videos that display discrimination based on race, gender, sexual orientation, age, disability, appearance (e.g., obesity), or religion.
- *Animal Cruelty.* Videos that depict the abuse or mistreatment of animals.
- *Dangerous Activities.* Videos that include dangerous content such as drug use, criminal behavior, bullying, terrorism, rumor propagation, incitement, or misinformation.
- *Physical Violence.* Videos containing acts of physical violence against individuals, including fighting, severe injuries, bleeding, self-harm, or torture.
- *Obscenities.* Videos that contain explicit language, sexual behavior, or suggestive content.
- *Shocking Content.* Videos that include startling or fear-inducing elements such as gunshots, explosions, or jump scares.

In addition to harmful content detection, videos are also evaluated based on their quality:

- *Confusing:* Videos that are incomplete or otherwise difficult to understand.
- *Low Resolution:* Videos with a level of clarity that makes it challenging to discern the content.

**Chaplin video segmentation.** We selected 62 silent films by Charlie Chaplin and hired annotators to meticulously review each film, manually recording humorous moments to ensure each mime clip illustrates a whole mime through a single event or multi events. And we removed videos where both the reason for the humor and the action were repetitive (e.g. humor arising from a comical action due to inflexibility, such as failing to position a ladder properly) to ensure the quality and consistency of the videos and their annotations.

**Speech reliance minimization.** To ensure reliable identification of humorous content, we instructed two annotators to independently review each video and confirm the presence of clear humor. Each annotator was also instructed to review each video and label whether humor was primarily conveyed through visual cues and could be understood independently of speech. Only videos for which both annotators agreed were retained for the final dataset. We further employed Whisper (Radford et al., 2023), a performant speech-to-text model, to transcribe audio. Since Whisper transcribes filler sounds (e.g., "uh," "hmm") and other minimal utterances, we excluded any videos where the transcribed text exceeded 10 characters. Additionally, videos containing non-English speech were retained but muted, removing dependence on linguistic cues.

## B.2 ANNOTATION

**Annotator training.** We provided appropriate annotation training for crowdworkers, offering detailed explanations of the annotation platform's usage and the annotation guidelines for different tasks. Additionally, we supplied an annotation manual (Figure 6) and corresponding instructional videos, which included specific descriptions and examples of the annotation requirements for crowdworkers to consult at any time during the annotation process.

**Qualification.** The recruited crowdworkers were mainly from China, all possessing at least an undergraduate education and with English background. Before formal annotation began, we conducted training sessions and a qualification review. During the qualification stage, crowdworkers were required to annotate 15 video samples. We manually reviewed their results and assigned scores based on the annotation guidelines. Ultimately, we selected eight qualified annotators.

For the annotation process, we adopted a three-person collaborative annotation scheme, ensuring that each data entry underwent three rounds of annotation. First, an annotator performed the initial annotation. Next, a second annotator reviewed and supplemented the annotation. Finally, a third annotator reviewed and further refined the previous two rounds of annotations. The annotators rotated through these three roles, and each annotation round was tracked to ensure that the three rounds for each data entry were completed by different annotators. For humor rating and video captions, annotators were required to independently provide their own answers. For the remaining annotation tasks, when the second and third annotators reviewed and modified the previous annotations, they were required to submit a new annotation if they identified any issues. If a specific annotation issue was modified in all three rounds for a given video, we conducted a final review to assess the validity of the annotation results.

**Consensus.** For annotations like humor explanation and video description, the second and third annotators reviewed and modified previous annotations to ensure consistency. We employ Krippendorff's alpha (Krippendorff, 2011) to assess the annotators' consensus on the humor evaluation, using 'Low,' 'Medium,' and 'High' to indicate the strength of the consensus. In our dataset, more than 90% of the annotated data demonstrated a 'High' consensus, while only 0.4% showed a 'Low' consensus.

## B.3 COPYRIGHT & LICENSE

We respect the copyright of each video. We have already emailed Charlie Chaplin's copyright holders regarding copyright issues related to Chaplin clips, and v-HUB is only used for academic research. Commercial use in any form is prohibited. The copyright of all videos belongs to the video owners. Without prior approval, you cannot distribute, publish, copy, disseminate, or modify v-HUB in whole or in part. You must strictly comply with the above restrictions.

## C DATA STATISTICS

**Duration.** All videos in our final dataset are restricted to a duration of 5–60 seconds, with the majority concentrated within 30 seconds (see Figure 7a). This design ensures that humor is self-contained, sufficiently nuanced, and compatible with the context length limits of most MLLMs.

**Diversity.** To show our dataset contains a variety of humor, we follow Buijzen & Valkenburg (2004) to categorize humor into five categories (see Figure 7b): Slapstick humor, Clownish humor, Surprise, Irony, Misunderstanding and Others (e.g. Parody, Miscellaneous, Satire). And we follow House et al. (2004); Ronen & Shenkar (2013) to categorize cultural background into six categories (see Figure 7c): North American, Western European, East Asian, Latin American, South & Southeast Asian, Eastern European, and Others (e.g. Middle Eastern and North African)

**Visual-centric.** Our dataset is predominantly visual centric, with 99% of videos relying primarily on visual cues. Specifically, per the four groups defined in Section 2.4: *Visual*, *Visual+Text*, *Visual+Audio*, and *Visual+Autio+Text*, 600 videos (63%) fall into the category *Visual*, meaning humor is entirely derived from facial expressions, object interactions, or visual effects without reliance on text or sound effects. 214 videos (22%) integrate audio, indicating that while humor remains visually driven, auditory elements such as background music or sound effects enhance the comedic impact. 92

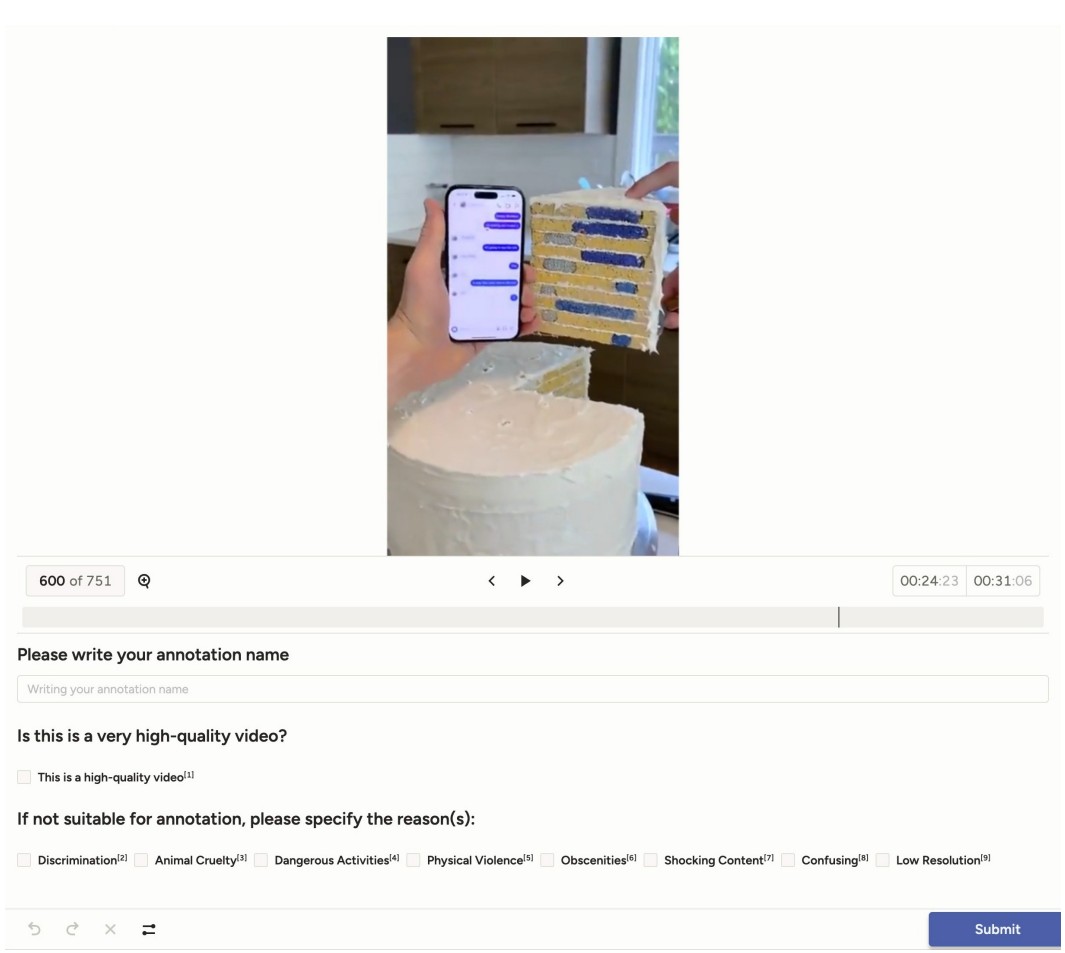

Figure 4: Interface for the Harmful Content Detection HIT.

Figure 5: Interface for HIT.

# Annotation Manual

**Please write your annotation name:**

**1. Please rate how humorous you find the video:**

☐ Very Humorous

☐ Somewhat Humorous

☐ Little bit Humorous

☐ Can't find any Humorous point

For this question, please rate the humor level based on your first impression of the video.

**2. Does the video contain any speech subtitles or visual text?**

☐ Speech Subtitles Exists

☐ Visual Text Exist

For this question please select whether subtitles and text information can be seen in the video.

1. Details of options:

- **Speech Subtitles**: Subtitles at the bottom of the video transcribing dialogues, narrations, and other speech, generally consistent with the spoken words.
- **Visual Text**: Text visible in the video apart from subtitles, including added text in the video, text on objects, etc.

2. Use clear recognition by the human eye as the standard. If the content is difficult to see clearly, you do not need to select it.

Annotation Manual                                                    1

---

**Choose one Question between Question 3, 4 to Answer:**

**3. Use your imagination to write a caption for the video that adds a new point of humor.**

This question ask you to add a caption to the video to increase its humor.

1. The video caption must be humorous only when combined with the video; the humor should not be understood by reading the text or watching the video alone.

2. Please ensure it is related to the video content.

3. The caption should not exceed one sentence.

**4. Pretend you're posting this video online. What caption will you write to make it more entertaining for viewers?**

This question ask you to write a caption for the video from the perspective of the video publisher. The caption should be connected to the video.

1. The caption should emphasize or enhance the humor of the video as much as possible. As the video publisher, you want to attract viewers.

2. Please ensure it is related to the video content.

3. The caption should not exceed one sentence.

**5. Which aspects of the video contributed to understanding humor?**

☐ Visual - Human
☐ Visual - Others
☐ Visual Effects
☐ Visual Text
☐ Speech Subtitles
☐ Speech

Annotation Manual                                                    2

---

☐ Sound Effects

**This question ask you to based on your understanding of the humor in the video, select the sources of information necessary for understanding the video and its humor.**

1. If a certain aspect appears in the video but does not help in understanding the video, you do not need to select it.

2. The following are detailed explanations of the above categories:

- **Visual - Human**: Information about human activities seen, including human expressions, actions, etc.
- **Visual - Others**: Information seen other than human activities, including objects, backgrounds, creatures, etc.
- **Visual Effects**: Post-production effects in the video, including special effects, filters, editing, etc.
- **Visual Text**: Text seen in the video apart from subtitles, including added text in the video, text on objects, etc.
- **Speech Subtitles**: Subtitles at the bottom of the video transcribing dialogues, narrations, and other speech, generally consistent with the spoken words.
- **Sound Effects**: Audio information heard, including music, sound effects, meaningless shouts, exclamations, etc.
- **Speech**: Spoken words heard, including dialogues, narrations, etc.

**6. Please describe the video in a direct way. (Detailing the main actions, people, and events without interpretation)**

Please describe what is happening in the video based only on what you see, including all the details necessary to understand the humor.

1. Please only describe the things that appear in the footage; do not make any inferential descriptions.

2. You may consider including:

Annotation Manual                                                    3

---

- Where does the video take place? Are there any changes in the scene?
- Who appears in the video, and what are they doing?
- What objects in the video need attention?
- What are the expressions of the people in the video?

**7. Please provide keywords / phrases representing background knowledge not shown in the video that an AI would need to understand the humor.**

The question requires you to analyze the background knowledge necessary for understanding the video.

1. Please include only knowledge that cannot be directly obtained by watching the video. If there is none, you do not need to answer.

2. Please ensure it is directly related to understanding the humor.

3. Please be as specific as possible. For example: "5G" is better than "Networks", "John F. Kennedy" is better than "US President".

**8. Using your answers from Questions 5 to 7, please explain why the video is humorous.**

The question requires you to explain the humor in the video.

1. Please answer based on your analysis of the video content in questions 5, 6, and 7. For example, if in question 5 you selected 'sound effect', explain why the sound effect makes people feel humor.

2. Please include all the humorous elements you can find, as thoroughly as possible.

Annotation Manual                                                    4

Figure 6: Interface for Annotation Manual for data annotation.

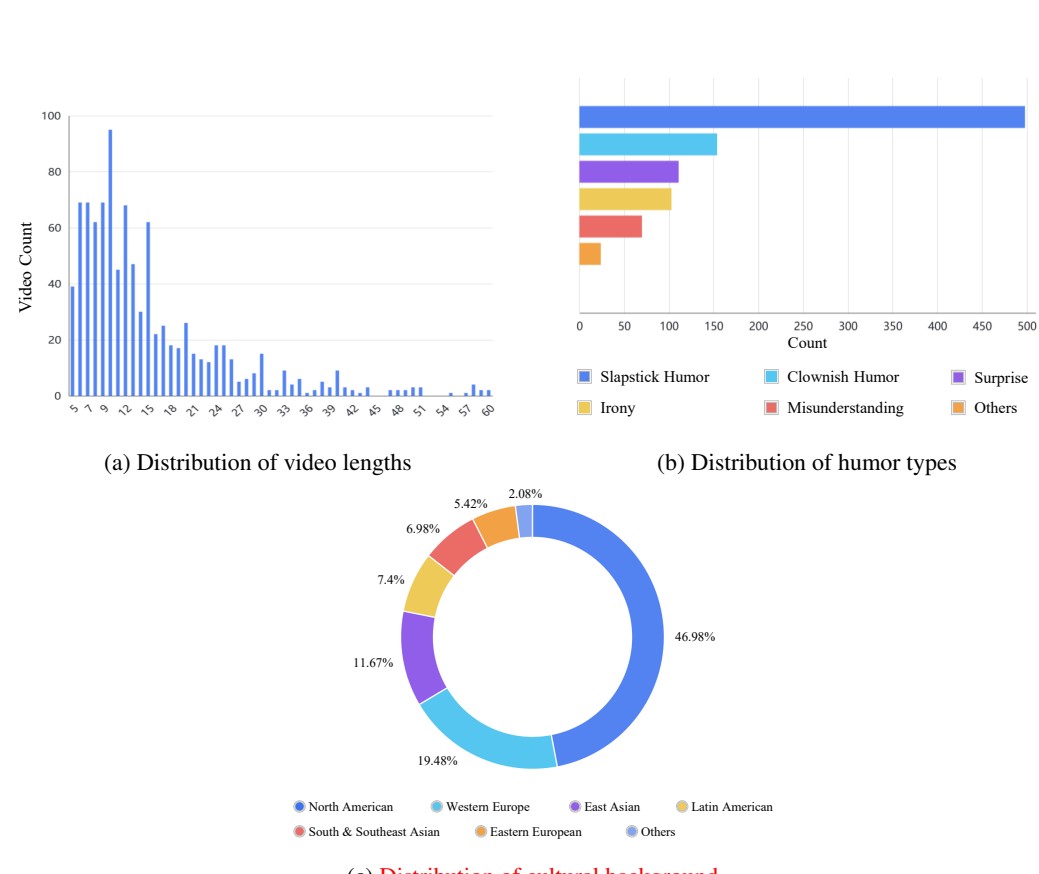

(a) Distribution of video lengths

(b) Distribution of humor types

(c) Distribution of cultural background

Figure 7: Data statistics: video length and humor type distributions.

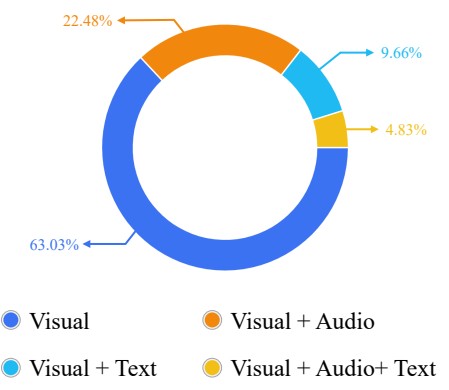

Figure 8: Video distribution over four groups.

Table 8: Evaluated Models.

| Models | #Parameter | Proprietary | Input Modality | | |
|---|---|---|---|---|---|
| | | | Text | Video | Video+Audio |
| Qwen2.5-VL | 72B | ✗ | ✓ | ✓ | ✗ |
| Qwen-2.5-Omni | 7B | ✗ | ✓ | ✓ | ✓ |
| Intern3.5-VL | 8B | ✗ | ✓ | ✓ | ✗ |
| MiniCPM2.6-o | 8B | ✗ | ✓ | ✓ | ✓ |
| Video-SALMONN-2 | 7B | ✗ | ✓ | ✓ | ✓ |
| GPT-4o | - | ✓ | ✓ | ✓ | ✗ |
| Gemini-2.5-flash | - | ✓ | ✓ | ✓ | ✓ |

videos (9%) are classified as *Visual+Text* indicating that video humor is extended through additional visual text. 46 videos (5%) combines the three modalities to deliver humor. The remaining 1% includes videos, in which speech plays a minor role but does not dominate the expression of humor.

# D  ADDITIONAL EXPERIMENTAL NOTES

## D.1  DETAILS OF GENERATE OPEN-ENDED QA PAIRS

We employed GPT-4o to generate QA pairs for each video, with the questions primarily covering temporal, descriptive, and causal aspects. The specific prompts used for QA generation are provided in Table 17. Subsequently, annotators manually reviewed and revised the QA pairs for each video to ensure their accuracy and quality.

## D.2  DETAILS OF EVALUATION METHODS

**AutoDQ.**  It evaluates the presence of humor-related events in the generated explanations (Wang et al., 2024a). It extracts key events from the model's output and compares them to ground truth (GT) annotations using entailment analysis. It provides three metrics: recall, precision, and F1 score, defined as:

- *Recall* measures the percentage of GT events entailed by the model-generated events.
- *Precision* measures the percentage of model-generated events that are entailed by GT events.
- *F1 Score* is the harmonic mean of precision and recall, serving as a balanced indicator that jointly reflects coverage and correctness.

The inclusion of AutoDQ allows us to evaluate factual correctness and event coverage, checking whether the explanations covers all the humorous points in the actual video content.

**Human evaluation.**  We randomly sampled 50 explanations generated by models for human evaluation. To ensure consistency in the evaluation criteria, we assigned one annotator to rate the humor explanations generated by models. The scores ranged from 0 to 100, and were subsequently normalized by a factor of 100, yielding final results within the range $[0, 1]$.

## D.3  BASELINE MODELS

To evaluate multimodal large language models' ability to understand video humor, we selected state-of-the-art models representing three distinct input modalities, as summarized in Table 8. Specifically, we include multimodal LLMs that process raw visual frames and text, and omni LLMs that integrate both text, video and audio signals. This set covers both public models (e.g., Qwen2.5-VL-72B, Intern3.5-VL) and proprietary models (e.g., Gemini-2.5-flash, GPT-4o), offering a broad perspective on current approaches. Each model is evaluated under all input conditions it can handle (see Section 4.1): for instance, omni-modal models can participate in the Text-Only, Video-Only, and Video+Audio groups, whereas multimodal models are tested exclusively with textual input and

Table 9: Human preference comparison of humor explanations across four model categories.

| Models | Proprietary | Type | Setting | | |
|---|---|---|---|---|---|
| | | | Text-Only | Video-Only | Video+Audio |
| Qwen2.5-VL-72B | ✖ | MLLM | 0.687 | 0.423 | – |
| Qwen2.5-Omni | ✖ | OmniLLM | 0.574 | 0.430 | 0.381 |
| GPT-4o | ✔ | MLLM | 0.654 | 0.576 | – |
| Gemini-2.5-flash | ✔ | OmniLLM | 0.651 | 0.546 | 0.566 |

raw visual frames. This setup allows us to isolate how each model category—multimodal and omni-modal—contributes to humor understanding across diverse input modalities.

## E    ADDITIONAL EXPERIMENTAL RESULTS

**Proprietary MLLMs show stronger resilience to multimodal inputs compared to public MLLMs.** The results in Table 9 indicate that current MLLMs rely heavily on linguistic cues to generate reasonable explanations, and struggle to effectively extract semantic information from raw visual or auditory signals. For example, Qwen2.5-VL-72B attains a preference score of 0.687 under Text-Only, significantly outperforming its Video-Only score of 0.423. Furthermore, although closed-source models demonstrate greater robustness under multimodal inputs, they still struggle to align visual and audio cues to enhance humor comprehension. For instance, Gemini-2.5-flash achieves 0.546 (Video-Only) and 0.566 (Video+Sound).

## F    THE USE OF LARGE LANGUAGE MODELS

In this work, we used LLMs as assistive tools in several stages of the research:

- **Dataset construction.** We initially employed GPT-4o to assist in generating candidate QA pairs and humor categories from video content. All outputs were subsequently reviewed and revised by human annotators to ensure correctness and quality.
- **Code assistance.** LLMs were used to help generate parts of the evaluation code, which were then verified and refined by the authors.
- **Writing support.** LLMs were used to write and polish some sentences in the paper for readability.

## G    CASE STUDY

We present our case studies in Figure 9.

## H    PROMPTS

We list our prompt in Tables 16 to 26

**Task 1: Caption Matching**

You will see five captions, pick the only one that best corresponds to the humor in the video. The correct caption may relate to the humor either directly or indirectly.

A. Is the moving-to-moving parabolic projection so accurate?
B. Very quick response
C. Probably the last time I played it.
D. No moral ethics!
E. The problem is that you don't use a white ball. ✅

**Model Answer:**

Text-Only: **E**          Video-Only: **E**          Video-Audio: **E**

**Task 2: Open-ended QA**

Q: What did the kitten do with the ball?
A: Caught it with its claws and put it into the hole.

**Model Answer:**

Text-Only: The kitten **caught the ball** with its claws and **put it into the hole**.

Video-Only: The kitten **didn't interact with the ball**; it was just present in the scene.

Video-Audio: **pushed it**.

(a) case study 1

**Task 1: Caption Matching**

You will see five captions, pick the only one that best corresponds to the humor in the video. The correct caption may relate to the humor either directly or indirectly.

A. Not a human at all
B. The big dumbbell of the skinny man
C. Where else are you hiding?
D. The ball goes in!
E. Watch this epic parking garage symphony featuring trombones and car doors—pure chaotic genius! ✅

**Model Answer:**

Text-Only: **E**          Video-Only: **B**          Video-Audio: **E**

**Task 2: Open-ended QA**

Q: What are the two people on the side of the car doing?
A: Holding and slamming the car doors.

**Model Answer:**

Text-Only: **Holding the car doors and slamming them** hard to make a sound in coordination with the horn, and then blowing the horn to join in the music.

Video-Only: **Talking**

Video-Audio: The two people on the side of the car are **pushing it**.

(b) case study 2

**Task 1: Caption Matching**

You will see five captions, pick the only one that best corresponds to the humor in the video. The correct caption may relate to the humor either directly or indirectly.

A. Really free ✅
B. Car window eating incident
C. Balenciaga is like this
D. Kick away
E. Putting this together is fitness

**Model Answer:**

Text-Only: **A**          Video-Only: **D**          Video-Audio: **A**

**Task 2: Open-ended QA**

Q: Why do few people use the free wifi?
A: The password is too long.

**Model Answer:**

Text-Only: The password is **too long**.

Video-Only: Wi-Fi is **not working**.

Video-Audio: The password is very long and **hard to remember**.

(c) case study 3

Figure 9: Three case studies illustrating our findings.

Table 10: Model performance on Humor Explanation.

| Models | Explanation | | | Open-ended QA |
|---|---|---|---|---|
| | Precision | Recall | F1 Score | BERTScore |
| *Text-Only* | | | | |
| Gemini-2.5-flash | 0.286 | 0.361 | 0.319 | 0.723 |
| video-SALMONN-2 | 0.262 | 0.330 | 0.292 | 0.639 |
| MiniCPM2.6-o | 0.278 | 0.322 | 0.298 | 0.543 |
| Qwen-2.5-Omini | 0.281 | 0.311 | 0.295 | 0.698 |
| Qwen-2.5-VL-72B | 0.303 | 0.348 | 0.324 | 0.738 |
| Intern3.5-VL | 0.298 | 0.361 | 0.326 | 0.689 |
| GPT-4o | 0.317 | 0.391 | 0.350 | 0.718 |
| *Video-Only* | | | | |
| Gemini-2.5-flash | 0.140 | 0.196 | 0.163 | 0.556 |
| video-SALMONN-2 | 0.044 | 0.042 | 0.043 | 0.525 |
| MiniCPM2.6-o | 0.104 | 0.096 | 0.100 | 0.452 |
| Qwen-2.5-Omini | 0.168 | 0.126 | 0.144 | 0.488 |
| Qwen-2.5-VL-72B | 0.126 | 0.144 | 0.135 | 0.550 |
| Intern3.5-VL | 0.118 | 0.116 | 0.117 | 0.542 |
| GPT-4o | 0.189 | 0.208 | 0.198 | 0.556 |
| *Video+Sound* | | | | |
| Gemini-2.5-flash | 0.143 | 0.196 | 0.165 | 0.554 |
| video-SALMONN-2 | 0.097 | 0.038 | 0.055 | 0.538 |
| MiniCPM2.6-o | 0.111 | 0.110 | 0.110 | 0.513 |
| Qwen-2.5-Omini | 0.128 | 0.113 | 0.120 | 0.529 |

Table 11: The impacts of audio (i.e., sound) and visual text on video humor understanding.

| Models | Sound contributing to humor | | | | | Sound not contributing to humor | | | | |
|---|---|---|---|---|---|---|---|---|---|---|
| | Explanation | | | | Open-ended QA | Explanation | | | | Open-ended QA |
| | BERTScore | Precision | Recall | F1 Score | BERTScore | BERTScore | Precision | Recall | F1 Score | BERTScore |
| *w/ visual text* | | | | | | | | | | |
| Gemini-2.5-flash | 0.566 | 0.255 | 0.271 | 0.262 | 0.561 | 0.564 | 0.191 | 0.204 | 0.197 | 0.573 |
| video-SALMONN-2 | 0.511 | 0.064 | 0.031 | 0.042 | 0.532 | 0.511 | 0.064 | 0.037 | 0.047 | 0.534 |
| MiniCPM2.6-o | 0.543 | 0.130 | 0.135 | 0.132 | 0.524 | 0.542 | 0.145 | 0.153 | 0.149 | 0.518 |
| Qwen-2.5-Omni | 0.552 | 0.184 | 0.158 | 0.170 | 0.532 | 0.541 | 0.166 | 0.144 | 0.154 | 0.547 |
| *w/o visual text* | | | | | | | | | | |
| Gemini-2.5-flash | 0.551 | 0.156 | 0.186 | 0.170 | 0.545 | 0.545 | 0.024 | 0.039 | 0.030 | 0.554 |
| video-SALMONN-2 | 0.509 | 0.064 | 0.039 | 0.049 | 0.529 | 0.505 | 0.092 | 0.034 | 0.050 | 0.542 |
| MiniCPM2.6-o | 0.532 | 0.113 | 0.109 | 0.111 | 0.519 | 0.518 | 0.033 | 0.047 | 0.039 | 0.509 |
| Qwen-2.5-Omni | 0.541 | 0.112 | 0.131 | 0.121 | 0.525 | 0.525 | 0.053 | 0.038 | 0.042 | 0.527 |

Table 12: Comparison between MLLMs and their base LLMs under the Text-Only setting.

| Models | Explanation | | | | | | Matching |
|---|---|---|---|---|---|---|---|
| | SentBERT | METEOR | BERTScore | Precision | Recall | F1 Score | Accuracy |
| Qwen2.5-VL-72B | 0.553 | 0.250 | 0.553 | 0.303 | 0.348 | 0.324 | 0.719 |
| Qwen2.5-72B | 0.546 | 0.245 | 0.581 | 0.297 | 0.348 | 0.321 | 0.646 |
| Qwen2.5-Omni-7B | 0.547 | 0.232 | 0.547 | 0.281 | 0.311 | 0.295 | 0.644 |
| Qwen2.5-7B | 0.568 | 0.241 | 0.573 | 0.268 | 0.342 | 0.301 | 0.522 |

Table 13: The impact of requiring background knowledge support on video humor understanding.

| Models | Explanation | | | Open-ended QA |
|---|---|---|---|---|
| | Precision | Recall | F1 Score | BERTScore |
| Gemini-2.5-flash | 0.188 | 0.199 | 0.194 | 0.550 |
| Qwen-2.5-Omni | 0.189 | 0.110 | 0.139 | 0.520 |
| MiniCPM2.6-o | 0.110 | 0.081 | 0.093 | 0.442 |
| video-SALMONN 2 | 0.043 | 0.018 | 0.026 | 0.486 |

Table 14: The impact of background knowledge on video humor understanding.

| Models | Explanation | | | Open-ended QA |
|---|---|---|---|---|
| | Precision | Recall | F1 Score | BERTScore |
| *w/ Background Knowledge* | | | | |
| video-SALMONN-2 | 0.116 | 0.105 | 0.112 | 0.535 |
| MiniCPM-2.6-o | 0.191 | 0.192 | 0.191 | 0.520 |
| Qwen-2.5-Omni | 0.196 | 0.161 | 0.177 | 0.556 |
| *w/o Background Knowledge* | | | | |
| video-SALMONN-2 | 0.078 | 0.013 | 0.023 | 0.528 |
| MiniCPM-2.6-o | 0.131 | 0.100 | 0.113 | 0.509 |
| Qwen-2.5-Omni | 0.145 | 0.115 | 0.129 | 0.526 |

Table 15: The impact of video era on video humor understanding.

| Models | Explanation | | | Open-ended QA |
|---|---|---|---|---|
| | Precision | Recall | F1 Score | BERTScore |
| *Last-Century Charlie Chaplin's Silent Films* | | | | |
| Gemini-2.5-flash | 0.118 | 0.145 | 0.130 | 0.545 |
| video-SALMONN-2 | 0.035 | 0.007 | 0.012 | 0.513 |
| MiniCPM2.6-o | 0.116 | 0.083 | 0.097 | 0.470 |
| Qwen-2.5-Omni | 0.153 | 0.070 | 0.096 | 0.493 |
| *Contemporary User-Generated Funny Video* | | | | |
| Gemini-2.5-flash | 0.160 | 0.219 | 0.185 | 0.560 |
| video-SALMONN-2 | 0.050 | 0.052 | 0.051 | 0.529 |
| MiniCPM2.6-o | 0.096 | 0.111 | 0.103 | 0.446 |
| Qwen-2.5-Omni | 0.174 | 0.132 | 0.174 | 0.487 |

Table 16: Prompt for writing captions of videos.

```
And I will provide a description of the video and a list of
descriptive captions that break down what happens in it.
Your task is to write a caption in one sentences from the video
creator's perspective -- something you would write to attract
viewers.
Requirements:
Please ensure it is related to the video content.
- Write as if you're sharing it with an audience (e.g., use
'this' or 'me' naturally).
Output format:
Caption:  <caption>
Video description:  {video_description}
Descriptive captions:  {descriptive_captions}
```

Table 17: Prompt for generate QA pairs.

```
These are frames from a video.
And you'll be given a description of a video and an explanation of
why it's humorous to watch.
Based on given information, generate a Video Reasoning QA pair,
try to make answer only as phrases.  Let's think step by step. \n
Additionally, classify this question into one of the following
categories using the concise definitions provided: \n
Descriptive question:  Involves factual details such as location
or count \n
Temporal question:  Involves time-related aspects (e.g., previous,
after) \n
Causal question:  Involves reasons or explanations (e.g., why,
how) \n\n
Example 1: \n
Description: \n
Two hands are stretched out, one hand holding KFC chicken nuggets
and the other hand holding seeds.  In the distance, a chicken runs
over, but the chicken prefers to eat the KFC chicken. \n
Explanation:  The chicken surprisingly likes to eat KFC chicken,
which is unexpected and a bit funny.  The man realizes something
is wrong and tries to push the chicken pieces away with his hand,
which adds to the humor with a sense of panic. \n\n
Question:  What does the man holding in his hand? \n
Answer:  KFC chicken nuggets and seeds. \n
Type:  Descriptive \n\n
Example 2: \n
Description:  A man poured red liquid into the water, and a group
of fish came to snatch the food.  Another man poured beer into the
water, and a group of men came to snatch the food like fish. \n
Explanation:  The portrait of people snatching food like fish
humorously reflects the attraction of beer to men, and the
connection between them is very funny. \n\n
Question:  After the man poured beer into the water, what
happened? \n
Answer:  A group of men came. \n
Type:  Temporal \n\n
Example 3: \n
Description:  A woman was lying on the handrail of an escalator
while moving down.  A man saw her, and lying on the handrail on
the other side, and as a result, there was no barrier on that
side, and he fell directly down the escalator. \n
Explanation:  The man tried to show off by imitating others, but
ended up falling hard, which made people find it funny. \n
Question:  Why does the man fall off on the other side of the
handrail? \n
Answer:  There was no barrier. \n
Type:  Causal \n\n
Output format: \n
Question:  <question> \n
Answer:  <answer> \n
Type:  <type> \n\n
Video Description:  {video_description} \n
Humor Explanation:  {humor_explanation} \n
```

Table 18: Prompt for video QA.

System: You are a helpful AI assistant. You can analyze
videos and answer questions about their content. Respond
with short and concise answers. Avoid using unpronounceable
punctuation or emojis.

User: These are frames from a video. Based on these frames,
answer the following question: {question} \n\n

Output format: \n
Answer: <answer> \n\n

Table 19: Prompt for video explanation.

System: You are a helpful AI assistant specialized in video
understanding and humor analysis. You can explain jokes
clearly and naturally based on video content and video
description. Please respond with short and concise answers.
Avoid using unpronounceable punctuation or emojis.

User: These are frames from a video. Your job is to explain
why the video is humorous in 2-3 sentences as if you were
explaning to a friend who doesn't get the joke yet. Respond
with a 2-3 sentence explanation of the joke and how it relates
to the video. \n\n

Output format: \n
Explanation: <answer> \n\n

Table 20: Prompt for video caption matching.

System: You are a helpful AI assistant. You can analyze
videos and answer questions about their content. Please only
output in the specified format. No extra text.

User: Along with the frames from the video. And {question} \n
Please respond with response with the option letter only. \n\n

Output format: \n

Table 21: Prompt for video with description QA.

System: You are a helpful AI assistant. You can analyze
videos and answer questions about their content. Respond
with short and concise answers. Avoid using unpronounceable
punctuation or emojis.

User: You'll be given a description of the video. Based on
this information, answer the following question: {question}
\n\n

Output format: \n
Answer: <answer> \n\n

Video Description: {video_description}

Table 22: Prompt for video with description explanation.

```
System:  You are a helpful AI assistant specialized in video
understanding and humor analysis.  You can explain jokes
clearly and naturally based on video content and video
description.  Please respond with short and concise answers.
Avoid using unpronounceable punctuation or emojis.

User:  You will also be given a description of the video.  Your
job is to explain why the video is humorous in 2-3 sentences as
if you were explaning to a friend who doesn't get the joke yet.
Respond with a 2-3 sentence explanation of the joke and how it
relates to the video. \n\n

Output format: \n
Explanation:  <answer> \n\n

Video Description:  {video_description}
```

Table 23: Prompt for video with description caption matching.

```
System:  You are a helpful AI assistant.  You can analyze
videos and answer questions about their content.  Please only
output in the specified format.  No extra text.

User:  You'll be given a description of the video.  And
{question}\n Please respond with response with the option
letter only.\n\n Output format:\n Answer:  <answer>\n\n Video
Description:  {video_description}
```

Table 24: Prompt for video with sound QA.

```
System:  You are a helpful AI assistant.  You can analyze
videos and answer questions about their content.  Respond
with short and concise answers.  Avoid using unpronounceable
punctuation or emojis.

User:  Here's a humorous video.  Based on the its visual and
audio information, answer the following question:  {question}
\n\n

Output format: \n
Answer:  <answer> \n\n
```

Table 25: Video with sound explanation.

```
System:  You are a helpful AI assistant specialized in video
understanding and humor analysis.  You can explain jokes
clearly and naturally based on video content and video
description.  Please respond with short and concise answers.
Avoid using unpronouncable punctuation or emojis.

User:  Here's a humorous video.  Your job is to explain why
the video is humorous in 2-3 sentences as if you were explaning
to a friend who doesn't get the joke yet.  Respond with a 2-3
sentence explanation of the joke and how it relates to the
video. \n\n

Output format: \n
Explanation:  <answer> \n\n
```

Table 26: Video with sound caption matching

```
System:  You are a helpful AI assistant.  You can analyze
videos and answer questions about their content.  Please only
output in the specified format.  No extra text.

User:  Along with visual and audio information in the video.
And {question} \n
Please respond with response with the option letter only. \n\n

Output format: \n
Answer:  <answer> \n\n
```

