# OpenReview forum: "v-HUB: A Visual-Centric Humor Understanding Benchmark for Video LLMs"
_ICLR.cc/2026/Conference — ICLR 2026 Conference Withdrawn Submission_

### Official Review · Reviewer_mMpg · 2025-10-27

**Soundness:** 2
**Presentation:** 2
**Contribution:** 2
**Rating:** 2
**Confidence:** 4

**Summary:**

The paper introduces v-HUB, a visual-centric benchmark for evaluating multimodal large language models (MLLMs) on video humor understanding, addressing the gap in existing benchmarks that either rely on spoken language or require both video and linguistic cues to comprehend humor. The key findings of the paper include: MLLMs heavily rely on linguistic cues, struggle with active humor discovery, benefit slightly from audio + video, and perform worse on historically distant silent films than contemporary videos.

**Strengths:**

1. The paper performs a comprehensive comparison of existing datasets (e.g., NYCC, MUStARD, SMILE, ExFunTube) in the area of multimodal humor understanding and systematically diagnoses their inherent limitations.
2. This paper designs three complementary tasks: Caption Matching, Humor Explanation, Open-ended QA. They cover diverse cognitive dimensions of humor understanding—from deep video-text alignment to active humor discovery and general video reasoning.

**Weaknesses:**

1. The dataset scale is relatively limited, with only 900+ videos included. This small sample size may restrict the evaluation breadth. For instance, it fails to fully cover humorous scenarios across diverse cultural backgrounds or visual styles, making it difficult to disentangle whether the observed MLLM underperformance stems from inherent capability gaps or insufficient dataset coverage.
2. Additionally, this dataset may lack of diversity (e.g., different types of humor, different topics of videos).  This lack of transparent cultural diversity validation restricts the generalizability of v-HUB to MLLMs evaluated on cross-cultural humor understanding tasks.
3. The above limited scale and diversity of historical humor samples undermines the statistical robustness of the claim regarding MLLMs’ poorer comprehension of historical versus contemporary humor, as the conclusion may be biased by small-sample variation rather than genuine model limitations.
4. The paper lacks explicit documentation of the parameter sizes for most evaluated models (e.g., Video-SALMONN-2, MiniCPM-2.6-o) and omits controlled experiments to isolate the impacts of model parameters versus architecture. While a few models (e.g., Qwen2.5-VL-72B) have parameter sizes specified, the absence of comparative tests between models with identical architectures but different parameter scales (or vice versa) makes it impossible to attribute performance differences to either pre-trained knowledge storage gaps (due to parameter size) or architectural design nuances (e.g., visual encoders, multimodal fusion modules, the pre-training method of different model). This ambiguity weakens the depth of analysis into MLLMs’ humor understanding limitations.
5. The evaluation framework does not include comparative experiments on performance-boosting methods such as few-shot prompting, chain-of-thought (CoT), or fine-tuning. The paper cannot assess whether MLLMs’ visual-centric humor understanding capabilities can be activated or enhanced via these approaches.
6. While the paper mentions that some qualified annotators conducted three rounds of annotation for each video, it does not report quantitative metrics (e.g., Cohen’s Kappa coefficient, Fleiss’ Kappa) to measure inter-annotator agreement.
7. This work mainly uses traditional NLG automatic metrics to evaluate the ability of LLMs. Personally, I feel that these metrics are no longer sufficient to indicate whether the generated text conforms to human preferences.

**Questions:**

1.V-HUB only contains 900+ videos and is sourced by a single X account and silent flims from Charlie Chaplin, how do you address the concern that the small and undiverse dataset scale may limit the evaluation breadth, diversity and statistical robustness of the claim about MLLMs’ poorer comprehension of historical versus contemporary humor? What specific strategies could be adopted to expand the dataset to better disentangle model inherent capability gaps from dataset coverage limitations?
2.The paper claims v-HUB is "visual-centric" (99% of videos rely on visual cues) but provides no operationalized threshold for "visual reliance". How were annotators instructed to distinguish "visual as primary" from "visual as supplementary" when audio/text elements existed? Why was this definition not quantified, and how might ambiguity here affect the interpretability of experimental results?
3.Most evaluated models (e.g., Video-SALMONN-2, MiniCPM-2.6-o) lack explicit parameter size documentation, and no controlled experiments were designed to isolate the effects of parameter scale versus architectural design on performance. Why were these critical details and experiments omitted? What experimental design (e.g., comparing models with identical architectures but different parameter sizes) would help attribute performance differences to pre-trained knowledge storage or architectural nuances?
4.Lack of experiments on performance-boosting methods (e.g., few-shot prompting, chain-of-thought, fine-tuning) were included in the evaluation. Why were these approaches not tested, and how does this omission limit insights into MLLMs’ potential for visual-centric humor understanding?

---

> ### Author Response · Authors · 2025-11-26
> **Response to Reviewer mMpg**
>
> Thank you for your comment.
>
> 1.
> >  Weakness 1: The dataset scale is relatively limited, with only 900+ videos included. This small sample size may restrict the evaluation breadth.
>
> > Weakness 2:This dataset may lack of diversity (e.g., different types of humor, different topics of videos).
>
> **Answer:**
>
> We have strived to ensure that the dataset encompasses a wide range of cultural contexts and humor styles:
>
> **Multicultural**: Our dataset includes humorous videos from multiple regions, such as North American, Western European, Latin American, and East Asian (see details in Figure 7(c)).
>
> **Multiple humor styles**: Our dataset covers various humor styles, including slapstick humor, clownish humor, surprise, irony, and misunderstanding (see details in Figure 7(b)).
>
> **Different Sources**: We curate a set of visual-centric humorous videos from two complementary sources: Charlie Chaplin’s silent films and user-generated short funny videos from the Internet.
>
> ---
> 2.
> > Weakness 2: The above limited scale and diversity of historical humor samples undermine the statistical robustness of the claim regarding MLLMs’ poorer comprehension of historical versus contemporary humor, as the conclusion may be biased by small-sample variation rather than genuine model limitations.
>
> **Answer:**
>
> Benchmarks proposed by Loakman et al. [1] and Hyun et al. [2] show that datasets containing multiple classes with fewer than 1,000 samples can still reveal performance differences in LLM/VLM. Accordingly, v-HUB can reliably conclude that MLLMs face greater difficulty in understanding historically distant videos.
>
> And to ensure the scale and diversity of the historical humor samples, the clips were collected from 62 of Charlie Chaplin’s films, covering 75% of his complete filmography (82 films), encompassing a wide range of themes including everyday comedy, workplace satire, social and family comedy, chaotic action comedy, social class satire, and meta-cinematic works, demonstrating a rich diversity of content.
>
> For Chaplin video segmentation, we selected 62 silent films by Charlie Chaplin and hired annotators to meticulously review each film, manually recording humorous moments to ensure each mime clip illustrates a whole mime through a single event or multiple events. And we removed videos where both the reason for the humor and the action were repetitive (e.g. humor arising from a comical action due to inflexibility, such as failing to position a ladder properly) to ensure the quality and consistency of the videos and their annotations.
>
> [1] Loakman T, Thorne W, Lin C. Comparing apples to oranges: A dataset & analysis of llm humour understanding from traditional puns to topical jokes[J]. arXiv preprint arXiv:2507.13335, 2025.
>
> [2] Hyun L, Sung-Bin K, Han S, et al. Smile: Multimodal dataset for understanding laughter in video with language models[C]//Findings of the Association for Computational Linguistics: NAACL 2024. 2024: 1149-1167.
>
> ---
> 3.
> > Weakness 4: The paper lacks explicit documentation of the parameter sizes for most evaluated models (e.g., Video-SALMONN-2, MiniCPM-2.6-o) and omits controlled experiments to isolate the impacts of model parameters versus architecture.
>
> **Answer:**
>
> The parameters of the evaluation models Video-SALMONN-2, MiniCPM-2.6-o and Intern3.5-VL are 7B, 8B, and 8B, respectively (see Table 8). A control experiment was designed to isolate the impact of parameter size and architecture design on performance. We compared the performance of the Qwen2.5-Omni-7B and Qwen2.5-7B models, as well as the Qwen2.5-VL-72B and Qwen2.5-72B, under text-only conditions, and concluded that multimodal training can confer advantages even when only textual descriptions are available.
>
> ---
> 4.
> > Weakness 5: The evaluation framework does not include comparative experiments on performance-boosting methods such as few-shot prompting, chain-of-thought (CoT), or fine-tuning. The paper cannot assess whether MLLMs’ visual-centric humor understanding capabilities can be activated or enhanced via these approaches.
>
> **Answer:**
>
> We referenced several existing benchmarks primarily targeting general video understanding tasks, such as MVBench, Video-MME, MLVU, and LVBench. These all use zero-shot as the default evaluation mode, without additional CoT, few-shot, or fine-tuning. Benchmarks aim to measure a model's intrinsic capability. v-HUB aims to evaluate the model's true visual humor understanding ability, rather than to prove that the model can improve after optimization. Therefore, our evaluation does not require such experiments.

---

> ### Author Response · Authors · 2025-11-26
> **Response to Reviewer mMpg**
>
> 5.
> > Weakness 6: While the paper mentions that some qualified annotators conducted three rounds of annotation for each video, it does not report quantitative metrics (e.g., Cohen’s Kappa coefficient, Fleiss’ Kappa) to measure inter-annotator agreement.
>
> **Answer:** For annotations like humor explanation and video description, the second and third annotators reviewed and modified previous annotations to ensure consistency. We employ Krippendorff’s alpha to assess the annotators' consensus on the humor evaluation, using 'Low,' 'Medium,' and 'High' to indicate the strength of the consensus. In our dataset, more than 90% of the annotated data demonstrated a 'High' consensus, while only 0.4% showed a 'Low' consensus.
>
> Table 1: Consensus Evaluation.
> | Consensus| Count (Percentage) |
> |-----------------|--------------------|
> | Low             | 4 (0.4%)           |
> | Medium          | 90 (9.4%)          |
> | High            | 866 (90.2%)        |
>
> ---
> 6.
> > Weakness 7: This work mainly uses traditional NLG automatic metrics to evaluate the ability of LLMs. Personally, I feel that these metrics are no longer sufficient to indicate whether the generated text conforms to human preferences.
>
> **Answer:** We acknowledge that using traditional NLG automatic metrics to evaluate the capabilities of LLM is a common practice, and many benchmarks also use these types of metrics for evaluation. For instance, ExFunTube[1], NYCC[2], and SMILE[3] employ conventional NLG metrics for evaluation, including SentBERT, METEOR, and BERTScore.
>
> We recognize the limitations of using only traditional NLG automatic metrics to evaluate the generated text; therefore, we also used the AutoDQ and human evaluation to evaluate the ability of LLMs in the humor explanation task (see details in Table 2, Table 9, and Appendix D.2). AutoDQ uses an extraction model to extract events from two video descriptions, then uses an entailment model to examine how many events extracted from one description are entailed by the other description. We use GPT-4o to implement both models.
>
> [1] Dayoon Ko, Sangho Lee, and Gunhee Kim. Can language models laugh at youtube short-form videos? arXiv preprint arXiv:2310.14159, 2023.
>
> [2] Jack Hessel, Ana Marasovic ́, Jena D Hwang, Lillian Lee, Jeff Da, Rowan Zellers, Robert Mankoff, and Yejin Choi. Do androids laugh at electric sheep? humor" understanding" benchmarks from the new yorker caption contest. arXiv preprint arXiv:2209.06293, 2022.
>
> [3] Lee Hyun, Kim Sung-Bin, Seungju Han, Youngjae Yu, and Tae-Hyun Oh. Smile: Multimodal dataset for understanding laughter in video with language models. arXiv preprint arXiv:2312.09818, 2023.

---

### Official Review · Reviewer_Qhvs · 2025-10-29

**Soundness:** 3
**Presentation:** 2
**Contribution:** 2
**Rating:** 2
**Confidence:** 4

**Summary:**

This work contributes a novel dataset named v-HUB as a video humor understanding benchmark. It consists of minimally verbal short videos, and labels for three target tasks of caption matching, humor explanation, and open-ended QA. With the new dataset, this work empirically reveals that existing MLLMs suffer from the issue of strong bias toward text over visuals.

**Strengths:**

1. This work contributes a novel dataset that can evaluate the MLLMs’ ability of humor understanding from visuals. This dataset can be a valuable resource for this line of research to the community.

**Weaknesses:**

1. The dataset novelty is limited.
- Table 1 summarizes the novelty of the proposed v-HUB over existing humor video datasets. However, it contains highly exaggerated arguments.
- Surely, v-Hub is more visual-centric than the other datasets, but it is an overclaimed argument that only v-Hub is O and the others are X, since it is a matter of degree not a binary O/X decision. For example, ExFunTube contains many visual-centric datapoints, although all datapoints would not.
- Also, another important novelty argument is that v-Hub supports three target tasks. However, once one has humor explanation text labels for humor videos, it is almost automatic to transform their label formats for those of caption matching and QA tasks. Thus, it is not a notable contribution.

2. The experimental findings are quite predictable with little surprising novelty.
- It is a well-known issue that current multimodal models (1) are highly biased toward text information over visual one, and (2) often ignore subtle information in the visuals. These phenomena have been observed in almost all multimodal tasks.
- Likewise, findings summarized in the text in bold in section 4.2-4.3 have little novel perspective on the multimodal task.

Based on the points in 1-2 (limited novelty), this work may be better fit to a second-tier venue rather than ICLR.

3. The ethics review is not possible for a reviewer.
- Often humor videos in social media are highly likely to contain sensitive, offensive, disturbing, or private content. However, this work does not provide reviewers with the means to assess such issues in the dataset, as only four examples in Fig.1 are viewable to a reviewer.
- The copyright issue on Charlie Chaplin’s Silent Films (one of the two main video sources) is not discussed.

**Questions:**

Please refer to the Weaknesses.

**Details Of Ethics Concerns:**

Please refer to the Weakness 3.

---

> ### Author Response · Authors · 2025-11-26
> **Response to Reviewer Qhvs**
>
> Thank you for your comment.
> 1.
> > Weakness 1: The dataset novelty is limited.
> · Table 1 contains highly exaggerated arguments.
> · Surely, v-Hub is more visual-centric than the other datasets, but it is an overclaimed argument that only v-Hub is O and the others are X, since it is a matter of degree not a binary O/X decision. For example, ExFunTube contains many visual-centric datapoints, although all datapoints would not.
> · Also, another important novelty argument is that v-Hub supports three target tasks. However, once one has humor explanation text labels for humor videos, it is almost automatic to transform their label formats for those of caption matching and QA tasks. Thus, it is not a notable contribution.
>
> Answer: Thanks for pointing out! We'd like to clarify as follows:
> - (a): Our work targets filling the gap in assessing the capacity of MLLMs for humor understanding. To this end, we introduced v-HUB, a visual-centric video humor understanding benchmark designed to gauge the performance of MLLMs.
> - (b) Visual-centric is a characteristic of videos where the humor is derived predominantly from facial expressions, object interactions, or visual effects without reliance on text. In v-HUB, 99% of videos rely primarily on visual cues. But ExFunTube's videos are funny from both verbal and visual elements, instead of funny from only one modality. The videos can be understood only when both the video and linguistic cues are present.
> - (c) We apologize for the inappropriate task categorization in Table 1, and the three target tasks also exist in benchmarks such as MVbench, ExFunTube, and NYCC. The format of humor explanations cannot be converted to the format required for caption matching task. Caption matching specifically requires creative captions that cannot be simply derived from humor explanations. Furthermore, if we can convert the humor explanation from humorous videos into the format for QA tasks, this effectively demonstrates the high quality of our annotation process.
>
> ---
> 2.
> > Weakness 2: The experimental findings are quite predictable with little surprising novelty.
>
> Answer: Previous studies have primarily focused on evaluating MLLMs on general video datasets and drawing conclusions based on these results. We conducted several experiments across a wide range of humor videos, and our findings focus on MLLM’s ability to understand humor, including:
> 1. Limited capability of MLLMs in inferring subtle humor and discovering humor (see Table 2)
> 2. Heavy reliance on linguistic cues for humor understanding (see Table 2).
> 3. Incorporating audio or visual text helps with video humor understanding (see Table 3).
> 4. Multimodal training can offer advantages even when only textual descriptions are available (see Table 7).
>
> ---
> 3.
> > Weakness 3: The ethics review is not possible for a reviewer.
> · Often humor videos in social media are highly likely to contain sensitive, offensive, disturbing, or private content. However, this work does not provide reviewers with the means to assess such issues in the dataset, as only four examples in Fig.1 are viewable to a reviewer. · The copyright issue on Charlie Chaplin’s Silent Films (one of the two main video sources) is not discussed.
>
> Answer:
> - (a) **During the Filtering**, we manually filtered out videos that contained potentially harmful content to ensure the safety and quality of the videos (See more details in Appendix B.1). We adhered to the safety objectives outlined in Thoppilan et al. (2022) and excluded videos that violated the established criteria, including: Discrimination, Animal Cruelty, Dangerous Acivities, Physical Violence, Obscenities, Shocking Content.
>
>   **During the Annotation**, we recruited eight annotators with broad cultural knowledge to interpret humor arising from various contexts, and provided appropriate annotation training for annotators, offering detailed explanations of the annotation platform’s usage and the annotation guidelines for different tasks.
>
> These measures helped minimize the likelihood of including videos containing sensitive, offensive, disturbing, or private content.
>
> ---
>
> - (b) We have added a Copyright & License section in the Appendix B.3 to address the relevant copyright issues. We respect the copyright of each video. We have already emailed Charlie Chaplin's copyright holders regarding copyright issues related to Chaplin clips, and v-HUB is only used for academic research. Commercial use in any form is prohibited. The copyright of all videos belongs to the video owners. Without prior approval, you cannot distribute, publish, copy, disseminate, or modify v-HUB in whole or in part. You must strictly comply with the above restrictions.

---

### Official Review · Reviewer_VXrF · 2025-10-30

**Soundness:** 3
**Presentation:** 4
**Contribution:** 1
**Rating:** 2
**Confidence:** 5

**Summary:**

This paper present a visual-centric video humor understanding benchmark, and give a comprehensive evaluation of many sota MLLMs.

**Strengths:**

1. The paper collects humor videos from both User-Generated Funny Videos and films, introduces new humor video sources.
2. Compare to existing video humor benchmarks, this paper evaluates a new generation of MLLM models (newer, lager, wider), demonstrating improvements in model capabilities and introducing new challenges in humor comprehension.

**Weaknesses:**

1. **Novelty**: The humor videoQA data type is already included in MVBench [1], Table 1 Action - Unexpected Action (What unexpected event contributes to the humor in the video?). The authors cited Mvbench in this paper but they didn't aware that their topic is already covered in it.
2. **Repetitive work**: Upon more precise tracing of the MVbench source, this paper's overall conceptual framework **nearly entirely overlaps with the HumorQA subset within FunQA [2]**.  This raises concerns about duplicate publication.
After a more precise comparison (FunQA vs v-HUB):
i) **Tasks**: Counter-intuitive timestamp - Caption matching, Title generation - Caption matching, Counter-intuitiveness reasoning - Humor explanation, FunQA-MCQA & Dialog subset - Open-ended QA.
ii) **Datasize**: 1,769' avg 7s vs. 960'.
iii) **Anno**: both annotate caption, description, explanation by human annotators.
iv) **Common result**: The models heavily relies on text cues and weak visual reasoning in humor comprehension.
v) **Eval metrics**: BLEURT, GPT4 vs. BERTScore, METEOR

This is more like a **coincidental repetition of research topics** (humor videoQA). However, give the existing weakness, even this paper introduces new benchmark and evaluation, **the omissions in its literature review are significant**, leading the authors to overestimate the novelty of their work and limiting the paper's potential for further advancement in MLLM humor comprehension.

[1] Li, K, et al. "Mvbench: A comprehensive multi-modal video understanding benchmark." Proceedings of the IEEE/CVF Conference on Computer Vision and Pattern Recognition. 2024.
[2] Xie, B., et al. "Funqa: Towards surprising video comprehension." European Conference on Computer Vision. Cham: Springer Nature Switzerland, 2024.

**Questions:**

1. Although existing research points are repetitive, authors can still leverage their existing data for in-depth studies, such as model training, ablation experiments involving more modalities, and thorough analysis of human cognitive patterns regarding humor.
2. Finally, **what are your thoughts and explanations regarding the overlap with HumorQA in FunQA**, and how do you plan to enhance  and reconstruct the value of your paper?

---

> ### Author Response · Authors · 2025-11-26
> **Response to Reviewer VXrF**
>
> Thank you for your comment.
>
> 1.
> > Weakness 1: Novelty: The humor videoQA data type is already included in MVBench [1], Table 1 Action - Unexpected Action (What unexpected event contributes to the humor in the video?). The authors cited Mvbench in this paper, but they didn't aware that their topic is already covered in it.
>
> **Answer**:
>
> MVbench consists of 20 complex video tasks, used to comprehensively evaluate the video understanding capabilities of existing MLLMs. v-HUB is designed to gauge and diagnose the capacity of MLLMs for humor understanding.
>
> In MVbench, the data type for the humor video QA is "Unexpected Action." This type corresponds to the "surprise" category in our dataset. For example, in Figure 1(a), a battery rolled against the conveyor belt's motion, instead of being delivered directly to the cashier as expected, forcing the cashier into an endless wait. This unexpected event contributes to the humor in the video. But the humor category includes more than just "Unexpected Action"; it also includes "Misunderstanding," "Irony," and "Clownish Humor." (We've already mentioned this point in the Diversity section of Appendix C.)
>
> Compared to MVbench, v-HUB focuses specifically on humor and covers a wider range of humor types, including Slapstick humor, Clownish humor, Surprise, Irony, Misunderstanding, Parody, Miscellaneous, and Satire (see details in Figure 7(b)). And v-HUB focuses on the auxiliary role of sound in humor, especially environmental sound, which is rich in common sense.

---

> ### Author Response · Authors · 2025-11-26
> **Response to Reviewer VXrF**
>
> 2.
> > Weakness 2: Repeating research topics (humor videoQA) and overestimating the novelty of the work.
>
> **Answer**:
>
> There are still several differences between v-HUB and the HumorQA subset within FunQA:
> - **Different Motivation and Research Question:** FunQA is designed to evaluate and enhance the depth of video reasoning based on counter-intuitive and surprising videos, to explore whether video models can understand humor or creativity in surprising videos. In contrast, our work targets a different motivation: filling the gap in assessing the capacity of MLLMs for humor understanding. To this end, we introduced v-HUB, a visual-centric video humor understanding benchmark designed to gauge the performance of MLLMs.
>
> ---
>
> - **Broader Humor Coverage**: HumorQA's data consists of surprising videos, and these are just one category (surprise) covered in our dataset. We have focused on humor and explored many other types of humor, including Slapstick humor, Clownish humor, Surprise, Irony, Misunderstanding, and Others (see details in Figure 7(b)).
>
> ---
>
> -  **Different Data Source:** FunQA consists of user-generated short surprise videos from the Internet, while v-HUB comprises humorous videos that are visual-centric and illustrate diverse humor. Charlie Chaplin's silent comedies are a classic example of visual-centric humor, but their themes and storytelling techniques are relatively limited. To enrich the diversity of humor in our dataset, we also included user-generated short videos from the Internet. Therefore, v-HUB contains complementary humor videos from two different domains.
>
> ---
>
> - **Modality Difference** Our data focuses on full modality, including audio like background music and sound effects, which is rich in common sense, and our findings also demonstrate that incorporating audio helps with video humor understanding. The FunQA dataset is designed to emphasize understanding 'fun' through the visual aspect of videos. However, the VideoChat in the paper can only convert the audio into captions for text encoding and then input them into the model. Therefore, it cannot handle videos containing background music and sound effects (e.g., the sound of wind, rain, and bells) that cannot be converted into captions, which are a crucial component of our dataset and what we aim to explore.
>
> ---
>
>
> - **More Systematic Evaluation:** FunQA emphasizes counter-intuitive reasoning, spatial-temporal reasoning, with its main evaluation tasks including timestamp localization, detailed description, and reasoning around counter-intuitiveness.
>
>   v-HUB aims to assess the capabilities of MLLMs to match and extend humorous content, reason about humor, and comprehend humor in context, with its three evaluation tasks: (1) Caption Matching, (2) Humor Explanation, and (3) Open-ended QA. To further examine whether MLLMs can understand humor in visual-centric videos through visual cues alone, as well as to evaluate the contribution of audio to humor comprehension, we consider three task settings: Text-Only, Video-Only, and Video+Audio.
>
> ---
>
> - **Different Insights and Discovery:**
>
>     FunQA's key findings include:
>     1. Timestamp localization is the most challenging task.
>     2. There is no clear winner across all tasks.
>     3. Performance varies greatly across different video types.
>
>     We conducted several experiments and made new findings (see Sections 4.2 and 4.3 of the main paper), including:
>     1. limited capability of MLLMs in inferring subtle humor and discovering humor (see Table 2).
>     2. Incorporating audio helps with video humor understanding (see Table 3).
>     3. MLLMs do not show a significant disadvantage in understanding videos that require background knowledge compared to those that do not (see Table 4).
>     4. Multimodal training can offer advantages even when only textual descriptions are available (see Table 7).
>
>     Therefore, FunQA primarily evaluates and enhances VLM's counter-intuitive reasoning ability, specifically its ability to understand humor or creativity in surprising videos. In contrast, v-HUB mainly explores the capability of MLLM to understand humor in videos with diverse styles and cultural backgrounds.

---

> > ### Comment · Reviewer_VXrF · 2025-11-27
> > **Response to Authors's rebuttal.**
> >
> > Thank you for your response.
> > 1. Missing related work and comparison with other benchmarks.
> >
> > When emphasizing the distinctiveness of your own work, it is still necessary to include the missing citations and dataset comparisons in the paper. As stated in Table 1 and the related work section, all prior benchmarks you listed are before 2023. In recent years, both the vision and NLP communities have explored humor video understanding, and these relevant benchmarks must be discussed; however, the necessary comparisons and citations (e.g., with MVBench, FunQA) have still not been added.
> >
> > 2. Motivation
> >
> > As you mentioned, previous work lacks audio analysis, and v-HUB carefully constructs audio-related tasks and analyses. However, your title and abstract repeatedly emphasize that v-HUB is “visual-centric.” Please maintain consistency in the positioning of the work.
> > That said, I appreciate the extensive audio-related analysis, and this is indeed an important and valuable component. I encourage you to strengthen its prominence within the main paper after the next revised version.
> >
> > 3. Data source
> >
> > As other reviewers have raised concerns regarding dataset quality, why not adopt widely-recognized community datasets and perform new annotations on top of them? If your Figure 7(b) shows that your humor-specific categorization cannot be achieved using existing datasets, then this justification needs to appear explicitly in the main text. Therefore, I strongly suggest moving Figure 7(a–c) into the main paper and clearly articulating the necessity of constructing a new dataset.
> >
> > 4. Copyright and Privacy
> >
> > Regarding copyright, I agree that the issue has been adequately addressed by your response. However, please pay attention to privacy protection: the faces shown in Figure 1(a) and Figure 9(b) must be blurred.
> >
> > I am happy to talk with authors and other reviewers and change my score if all concerns have been taken care of.

---

> > > ### Author Response · Authors · 2025-11-29
> > > **Response to Reviewer VXrF**
> > >
> > > Thank you for your thoughtful comments. We sincerely appreciate the time and effort you invested in reviewing our paper.

---

### Official Review · Reviewer_XGFv · 2025-11-01

**Soundness:** 3
**Presentation:** 4
**Contribution:** 3
**Rating:** 8
**Confidence:** 5

**Summary:**

This paper introduces V-HUB, a visual-centric video humor understanding benchmark that aims to address the limitations of prior works which primarily relied on text inclusion for humor comprehension.
V-HUB emphasizes visual understanding by providing rich multimodal annotations and a diverse set of video types.
It supports multiple tasks, such as humor explanation, caption matching, and open-ended question answering, making it a comprehensive resource for studying humor in videos.
The benchmark highlights the complex, multimodal nature of humor and the challenges of modeling subjectivity and cultural context in computational humor understanding.

**Strengths:**

**Well-defined problem statement and novelty:** The paper clearly defines its problem and contributes a visual-centric benchmark for humor understanding, addressing the gap in prior benchmarks that relied on natural language cues.

**High-quality benchmark:** Each video is accompanied by rich annotations, including closed captions, textual descriptions, explanations, and humor-related elements. These extensive annotations enable multifaceted evaluation of humor understanding across different dimensions.

**Support for multiple tasks:** The benchmark is applicable to various tasks related to video and humor understanding, such as humor explanation, caption matching, and open-ended question answering.

**Annotation methodology:** Annotations are collected using a dual-caption strategy, which explicitly addresses the subjectivity of humor by incorporating multiple perspectives.

**Diverse video sources:** V-HUB includes both silent and user-generated videos, spanning combinations of visual, visual+text, visual+audio, and video+text+audio modalities. This diversity ensures broad coverage of humor types and scenarios.

**Weaknesses:**

**Ambiguity between description and explanation:** The example in Figure 3 shows minimal distinction between a description and an explanation—the main difference being phrases like “which viewers found very humorous.” A deeper analysis is needed to clarify the conceptual and functional differences between these two annotation types to help readers understand the necessity and distinct role of each.

**Limited annotation granularity:** Some videos contain humor concentrated in short segments rather than throughout the entire clip.
Identifying these time segments could improve humor understanding.
Also, for videos with multiple independent humorous moments, the benchmark could benefit from segment-level annotations, ensuring that each humor instance is represented distinctly for more accurate analysis and model training.

**Insufficient detail about background knowledge (L421):** The paper states that background knowledge aids humor understanding but does not specify what this knowledge entails.
 Providing a concrete description or examples of such knowledge would enhance interpretability and reader comprehension.

**Questions:**

1. What do the authors see as the key differences between description and explanation?
 Were there any specific annotation guidelines to ensure they capture distinct aspects of humor?

2. Do all videos contain a single humorous moment, or are there cases with multiple independent humor instances?
 If so, does the annotation process capture these moments individually or collectively?

3. What exactly constitutes the background knowledge mentioned in L421? Could the authors provide examples or clarify how it was incorporated into the evaluation process?

---

> ### Author Response · Authors · 2025-11-26
> **Response to Reviewer XGFv**
>
> Thank you for your comment.
>
> 1.
> > Weakness 1: Ambiguity between description and explanation: The example in Figure 3 shows minimal distinction between a description and an explanation—the main difference being phrases like “which viewers found very humorous.” A deeper analysis is needed to clarify the conceptual and functional differences between these two annotation types to help readers understand the necessity and distinct role of each.
>
> Answer:
> - (a) Video description focuses on describing what is happening in the video based only on observable objects, actions, and expressions. In our annotation manual for video description, we require annotators to describe what is happening in the video based only on what they see, including all the details necessary to understand the humor, and not make any inferential descriptions. Annotators may consider the following points: Where does the video take place? Are there any changes in the scene? Who appears in the video, and what are they doing? What objects in the video need attention? What are the expressions of the people in the video?
> - (b) Humor explanation focuses on identifying all the humorous elements and explaining why these elements make people feel humor. In our annotation manual for humor explanation, we require that annotators include all the humorous elements they can find, as thoroughly as possible, and explain why these elements make people feel humor.
> - (c) For example, in Figure 1(b), the video description is that "A cow was tied to the bottom of a plane and transported to another location". And the humor explanation is that "The title asked how milkshakes are made in Switzerland, but viewers didn't expect the plane to be flying directly with cows in it, which is a very interesting way". For more details, please refer to our annotation manual shown in Figure 6.
>
> ---
> 2.
> > Weakness 2: Limited annotation granularity: Some videos contain humor concentrated in short segments rather than throughout the entire clip. Identifying these time segments could improve humor understanding. Also, for videos with multiple independent humorous moments, the benchmark could benefit from segment-level annotations, ensuring that each humor instance is represented distinctly for more accurate analysis and model training.
>
> Answer: Each video contains one humorous moment. (This has already been mentioned in the Duration of Section 2.2). During the filtering phase, we ensure that each video delivers a single and complete joke without relying on additional context and that the video length is as short as possible.
>
> ---
> 3.
> > Weakness 3: Insufficient detail about background knowledge (L421): The paper states that background knowledge aids humor understanding but does not specify what this knowledge entails. Providing a concrete description or examples of such knowledge would enhance interpretability and reader comprehension.
>
> Answer: Background knowledge is information that cannot be obtained from videos, but it helps AI understand humor. For example, in Figure 1(a), the video shows a man placed a battery on the conveyor belt, but it rolled against the belt’s motion, forcing the cashier into an endless wait. And the background knowledge of the physics of a rolling cylinder on a moving conveyor can help LLMs understand video content and humor. See our annotation manual in Figure 6 for more details.

---

> > ### Comment · Reviewer_XGFv · 2025-11-26
> >
> > Thank the authors for the response. All of my concerns have been addressed, and I will keep my score as it is.

---

> > > ### Author Response · Authors · 2025-11-26
> > >
> > > Thanks for your time and your support!

---

### Note · Authors · 2025-11-30

I have read and agree with the venue's withdrawal policy on behalf of myself and my co-authors.